# Robust cone-mediated signaling persists late into rod photoreceptor degeneration

**Miranda L Scalabrino[1], Mishek Thapa[1], Lindsey A Chew[1], Esther Zhang[1], Jason Xu[2], Alapakkam P Sampath[3], Jeannie Chen[4], Greg D Field[1]***

[1]Department of Neurobiology, Duke University School of Medicine, Durham, United States; [2]Department of Statistical Science, Duke University, Durham, United States; [3]Jules Stein Eye Institute, University of California, Los Angeles, Los Angeles, United States; [4]Zilkha Neurogenetics Institute, Keck School of Medicine, University of Southern California, Los Angeles, United States

**\*For correspondence:**
field@neuro.duke.edu

**Competing interest:** The authors declare that no competing interests exist.

**Abstract** Rod photoreceptor degeneration causes deterioration in the morphology and physiology of cone photoreceptors along with changes in retinal circuits. These changes could diminish visual signaling at cone-mediated light levels, thereby limiting the efficacy of treatments such as gene therapy for rescuing normal, cone-mediated vision. However, the impact of progressive rod death on cone-mediated signaling remains unclear. To investigate the fidelity of retinal ganglion cell (RGC) signaling throughout disease progression, we used a mouse model of rod degeneration (*Cngb1*^neo/neo^). Despite clear deterioration of cone morphology with rod death, cone-mediated signaling among RGCs remained surprisingly robust: spatiotemporal receptive fields changed little and the mutual information between stimuli and spiking responses was relatively constant. This relative stability held until nearly all rods had died and cones had completely lost well-formed outer segments. Interestingly, RGC information rates were higher and more stable for natural movies than checkerboard noise as degeneration progressed. The main change in RGC responses with photoreceptor degeneration was a decrease in response gain. These results suggest that gene therapies for rod degenerative diseases are likely to prolong cone-mediated vision even if there are changes to cone morphology and density.

## Editor's evaluation

This is an important study describing the decline of retinal function in a mouse model of slow photoreceptor degeneration. The authors present compelling evidence based on a characterization of functional changes across some RGC populations and information theory to assess the fidelity of the remaining. They show remarkable preservation of cone-driven ganglion cell light responses in advanced stages of a retinitis pigmentosa model when most rods have died and cone morphologies are dramatically altered. The results are presented clearly in the text and figures and are discussed in the context of previous studies on retinal degeneration.

## Introduction

Rod photoreceptor degeneration frequently leads to cone photoreceptor degeneration and death. Well before all rods have died, cones exhibit clear changes in morphology and physiology (*Hartong et al., 2006*; *Sahel et al., 2010*). This is likely due to the loss of trophic factors and structural support provided by rods to nearby cones (*Campochiaro and Mir, 2018*). The extent to which these changes in cone structure and function compromise the ability of the retina to reliably signal visual scenes at high light levels is not clear. One possibility is that rod degeneration has an immediate impact on

**eLife digest** Our sense of sight depends on the retina, a thin layer of cells at the back of each eye. Its job is to detect light using cells called photoreceptors, then send that information to the rest of the brain. The retina has two kinds of photoreceptors: rods (active in dim light) and cones (which detect colour and work in bright light). We rely heavily on cone cells for vision in our daily lives.

Retinitis pigmentosa is a progressive eye disease affecting photoreceptors. In the early stages of this disease, rods gradually die off. Next, cone cells start to die, inevitably resulting in blindness. There is currently no cure, although some experimental treatments are being developed that aim to prevent rod death or replace missing rod cells.

However, it is unclear if these therapies will be effective, because we do not fully understand how rod death affects cone cells – for example, whether or not it damages the cones irreversibly. Scalabrino et al. therefore set out to track how the signals that cones send to the brain changed over time during progression of the disease using genetically altered mice that reproduced the symptoms of retinitis pigmentosa.

In these mice, rod cells die off over several months, followed by complete loss of cones a few months later. Initial microscopy experiments looking at the shape and appearance of the cone cells revealed that the cones started looking abnormal long before all the rods died. Next, to determine if these unhealthy cones had stopped working, Scalabrino et al. measured the activity of the mice's retinal ganglion cells (RGCs) in bright light – in other words, when cones are normally active.

RGCs transmit signals from photoreceptors to the brain, like a 'telephone line' between our brains and eyes. Applying a technique called information theory – which was originally used to determine how efficiently signals travel down telephone lines – to these experiments revealed that the RGCs still sent high-quality visual information from the cones to the brain. This is was surprising because the cones appeared to be dying and were surrounded by dead rods.

This study sheds new light on the biological processes underpinning a devastating eye disease. The results suggest that treatments to restore vision could work even if given after a patient's cones start looking unhealthy, giving hope for the development of new therapies.

the ability of the retina to reliably signal visual scenes at cone-mediated light levels. Alternatively, homeostatic plasticity or redundancy in retinal circuitry may compensate for photoreceptor loss (*Care et al., 2020*; *Lee et al., 2021*; *Shen et al., 2020*). Such mechanisms could facilitate reliable signaling at the level of retinal output, despite deterioration in photoreceptor function. Identifying the extent to which changes in photoreceptor morphology impact retinal output will inform treatment timepoints for gene therapies aimed at halting rod loss to preserve cone-mediated vision.

To examine the impact of progressive rod loss on cone-mediated visual signaling, we used a mouse line that models a human form of retinitis pigmentosa (RP), a blinding disorder characterized by initial degeneration of rod photoreceptors that ultimately leads to cone degeneration. This mouse line, $Cngb1^{neo/neo}$, contains an insertion that interrupts translation of Cngb1, the beta subunit of the cyclic nucleotide-gated cation channel in rods (*Chen et al., 2010*; *Wang et al., 2019*). Without this subunit, normal channels fail to form, causing rods to be tonically hyperpolarized and ultimately resulting in rod death (*Hüttl et al., 2005*; *Zhang et al., 2009*). This degeneration is relatively slow, with approximately 30% rod loss at 1 month postnatal, complete loss of rods by 7 month postnatal, and complete cone loss by 8–9 months postnatal. Slow degeneration in this model provides a relatively large temporal window in which to assay changes in retinal signaling to increasing amounts of rod loss and accompanying changes in cone morphology and density. Slower forms of RP are also more common among the human population (*Grover et al., 1999*; *Hartong et al., 2006*), and thus slow degeneration models may be more therapeutically informative than animal models with relatively rapid photoreceptor degeneration (e.g., *rd1* and *rd10*).

To determine the changes in cone-mediated retinal signaling induced by progressive rod loss, we measured changes in the signaling of retinal ganglion cells (RGCs), the 'output' neurons of the retina. Deterioration in the fidelity of RGC signals captures net changes in retinal circuit function induced by rod degeneration: these RGC responses account for changes in retinal circuits that compensate or exacerbate deteriorating photoreceptor function. Large-scale multielectrode arrays (MEAs) were

used to measure the visual responses of hundreds of RGCs simultaneously in individual retinas while presenting a variety of visual stimuli: for example, checkerboard noise and natural movies. Three features of RGC signaling were specifically investigated. First, we measured when and how the spatio-temporal receptive fields (RFs) under cone-mediated conditions were altered by rod degeneration. These RFs are indicative of the visual features that are being signaled by the RGCs to the brain (*Chichilnisky, 2001*; *Keat et al., 2001*; *Yu et al., 2017*), and thus changes in these RFs represent changes in the kind of information being transmitted to the brain. Second, we probed how and when the spontaneous activity of RGCs was altered by rod degeneration. Many previous studies have noted the emergence of oscillations in the spontaneous activity of RGCs in animal models of RP. Such activity could disrupt the ability of the retina to encode stimuli. However, limited rodent models have been used to identify when these oscillations emerge relative to rod and cone photoreceptor death. These models result from different mutations, which cause distinct changes in rod physiology from those in *Cngb1^{neo/neo}* mice; when and how oscillations arise may be mutation specific and may not be ideal for capturing aspects of this heterogenous human disease (*Stasheff, 2008*; *Stasheff et al., 2011*). Third, we explored when and how rod degeneration impacts the fidelity of visual signals transmitted to the brain. Information theory was used to quantify changes in the fidelity of signaling naturalistic and artificial stimuli as a function of photoreceptor degeneration.

From 1 to 7 months of age in *Cngb1^{neo/neo}* mice, there were marked and clear changes in cone morphology and density resulting from rod degeneration. When assaying the spatiotemporal RFs of RGCs as a function of degeneration, there were subtle changes to the temporal RFs of RGCs as cone morphology changed. However, the spatiotemporal RFs of RGCs were remarkably stable until the latest stages of degeneration (5–7 months). The primary change to cone-mediated RGC responses was a decrease in response gain as photoreceptors were lost. Second, oscillations in the spontaneous activity of RGCs did not emerge in this mouse model until all light responses were

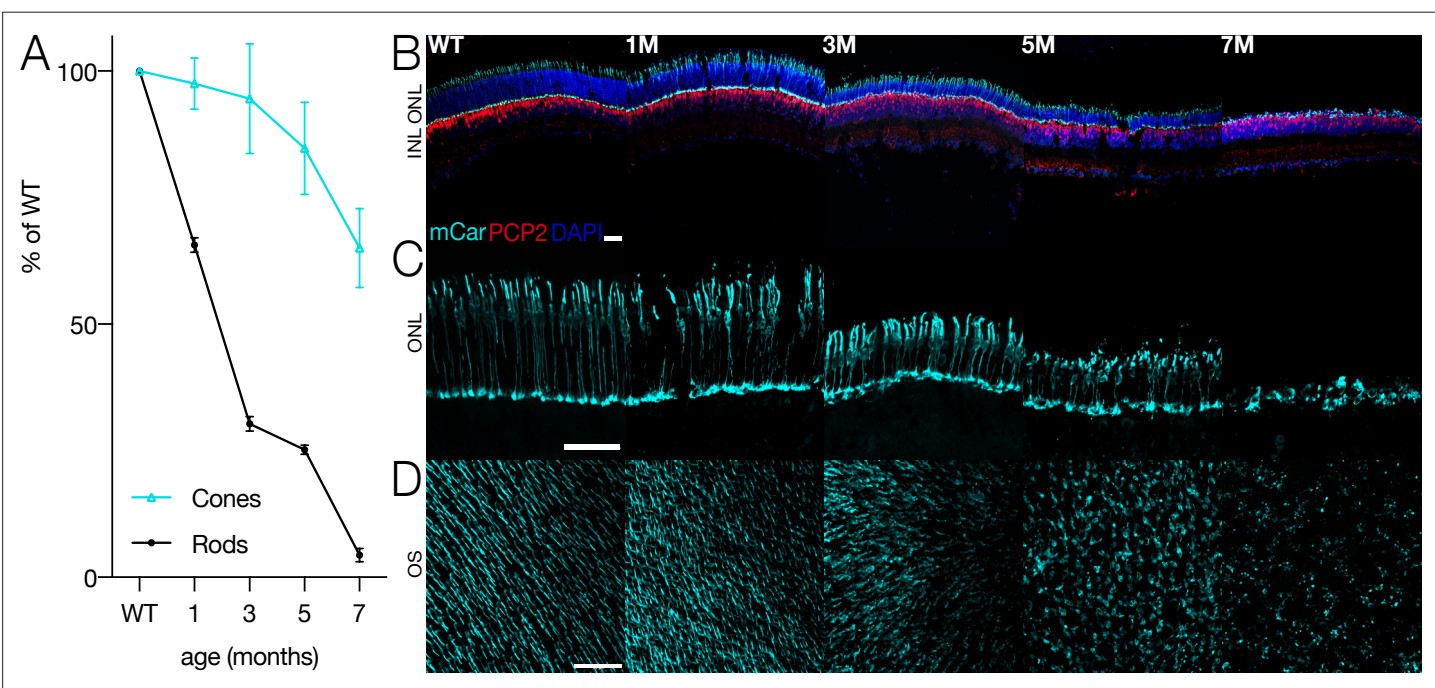

**Figure 1.** Cone morphology and density change with rod degeneration. (**A**) Estimated fraction of surviving rods (black) and cones (cyan) relative to wild-type (WT) densities from 1 to 7 months of age in *Cngb1^{neo/neo}* mice. (**B**) Immunofluorescence of retinal cross sections in WT and *Cngb1^{neo/neo}* mice (1–7 months). Cones (mCar) in cyan, bipolar cells (PCP2) in red, and nuclei (DAPI) in blue. (**C**) one morphologies imaged with cone arrestin (mCar) labeling from WT and *Cngb1^{neo/neo}* retinas. (**D**) whole-mount view of cone density and morphology in WT and *Cngb1^{neo/neo}* retinas. Scale bars: 50 µm. ONL: outer nuclear layer; INL: inner nuclear layer; OS: outer segments. Source files for (**A**) are available in *Figure 1—source data 1*.

The online version of this article includes the following source data and figure supplement(s) for figure 1:

**Source data 1.** Contains quantifications of rod and cone densities for *Figure 1A*.

**Figure supplement 1.** Nine-month *Cngb1^{neo/neo}* exhibit little-to-no immunolabeling for M-opsin.

eliminated (~9 months). This indicates that the onset of oscillatory activity can depend strongly on the genetic cause of RP, and thus may not be a concern for gene or cell therapy in at least some forms of RP. Third, as rods died and normal cone morphology deteriorated, the ability of RGCs to reliably signal the content of natural movies – under cone-mediated conditions – was remarkably robust until the latest stages of degeneration (~7 months; total rod loss with severe cone morphology changes and death). Thus, the fidelity of information transmission was relatively stable despite a reduction in response gain. This suggests that one or more mechanisms in the retina serve to compensate for deteriorating photoreceptors. Furthermore, these results suggest a broad therapeutic window in which to treat RP as cone-mediated visual signaling remains surprisingly robust despite clear changes in cone morphology and density.

## Results

### *Cngb1^{neo/neo}* cones slowly degenerate lagging rod death

Photoreceptors in *Cngb1^{neo/neo}* mice gradually die as a consequence of the *Cngb1* mutation (*Figure 1A*). While rods die first, shown by the shrinking outer nuclear layer (ONL) over time, cones are present until ~8 months (*Figure 1B*, *Figure 1—figure supplement 1*). However, cone structure begins to deteriorate at 3 months (*Figure 1C*, middle). Cone outer segments gradually shorten and eventually disappear prior to total cell death (*Figure 1D*). By 7 months, nearly all rods are lost and only cone cell bodies remain; clear outer segment structure is absent (*Figure 1C*, right). How does this deterioration in cone morphology and potential changes in cone function impact the ability of the retina to transmit visual information to the brain? Answering this question is particularly important given that humans primarily use cone vision for most tasks. One possibility is that cone-mediated signaling at the level of the RGCs is robust to changes in cone morphology. Alternatively, RGC signaling at cone-mediated light levels may rapidly deteriorate as rods are lost and cone morphology becomes abnormal.

### Identifying RGCs with space-time-separable receptive fields

To determine the impact of altered cone morphology and density due to rod death on retinal output, we began by measuring the RFs of RGCs from wild-type (WT) and *Cngb1^{neo/neo}* retinas using MEAs consisting of 512 or 519 electrodes (see 'Materials and methods'; *Anishchenko et al., 2010*; *Field et al., 2010*; *Litke et al., 2004*). The MEAs measured the spiking activity in 186–560 RGCs within individual samples of retina from 31 mice (see 'Materials and methods'). Animals were used at 1-month intervals from 1 to 7 months of age (postnatal). To estimate the spatiotemporal RFs of RGCs, we used checkerboard noise and computed the spike-triggered average stimulus (STA) for each RGC (*Chichilnisky, 2001*).

We measured RF structure at two light levels: a low mesopic level (100 Rh*/rod/s) at which cones are just beginning to be activated and a low photopic level (10,000 Rh*/rod/s). We chose the lower light level because *Cngb1^{neo/neo}* mice have severely compromised rod function even in surviving rods, thus we expected to see significant changes in visual signaling between WT and *Cngb1^{neo/neo}* (*Wang et al., 2019*). We chose the higher light level because it largely, if not completely, isolated cone-mediated signaling, thereby allowing us to examine the net impact of rod dysfunction and death on cone-mediated retinal output.

The STA estimates the linear component of each RF. We were particularly interested in changes to the spatial or temporal integration of visual input, which are estimated by changes in the spatial and temporal RFs, respectively. However, some RGC types (i.e., direction-selective RGCs) do not have an RF that can be decomposed into unique spatial or temporal filters (*Borghuis et al., 2008*; *Devries and Baylor, 1997*; *Vaney et al., 2012*), which makes it an ill-posed problem to separately analyze changes in spatial or temporal integration. Thus, we focused the analysis on RGCs with RFs that were well-approximated by a space-time-separable RF model: the outer product of two vectors, one describing the spatial RF and the other the temporal RF (*Figure 2*; see 'Materials and methods'). The time-dependent spiking activity of these RGCs in response to a checkerboard stimulus was also well-approximated by a linear–nonlinear Poisson model (*Figure 2—figure supplement 1A and B*), which is assumed when using the STA as an estimate of the RF (*Chichilnisky, 2001*).

To factorize the STA into the outer product of a spatial and a temporal filter, we used singular value decomposition (SVD) (*Golomb et al., 1994*; *Wolfe and Palmer, 1998*). SVD applied to a

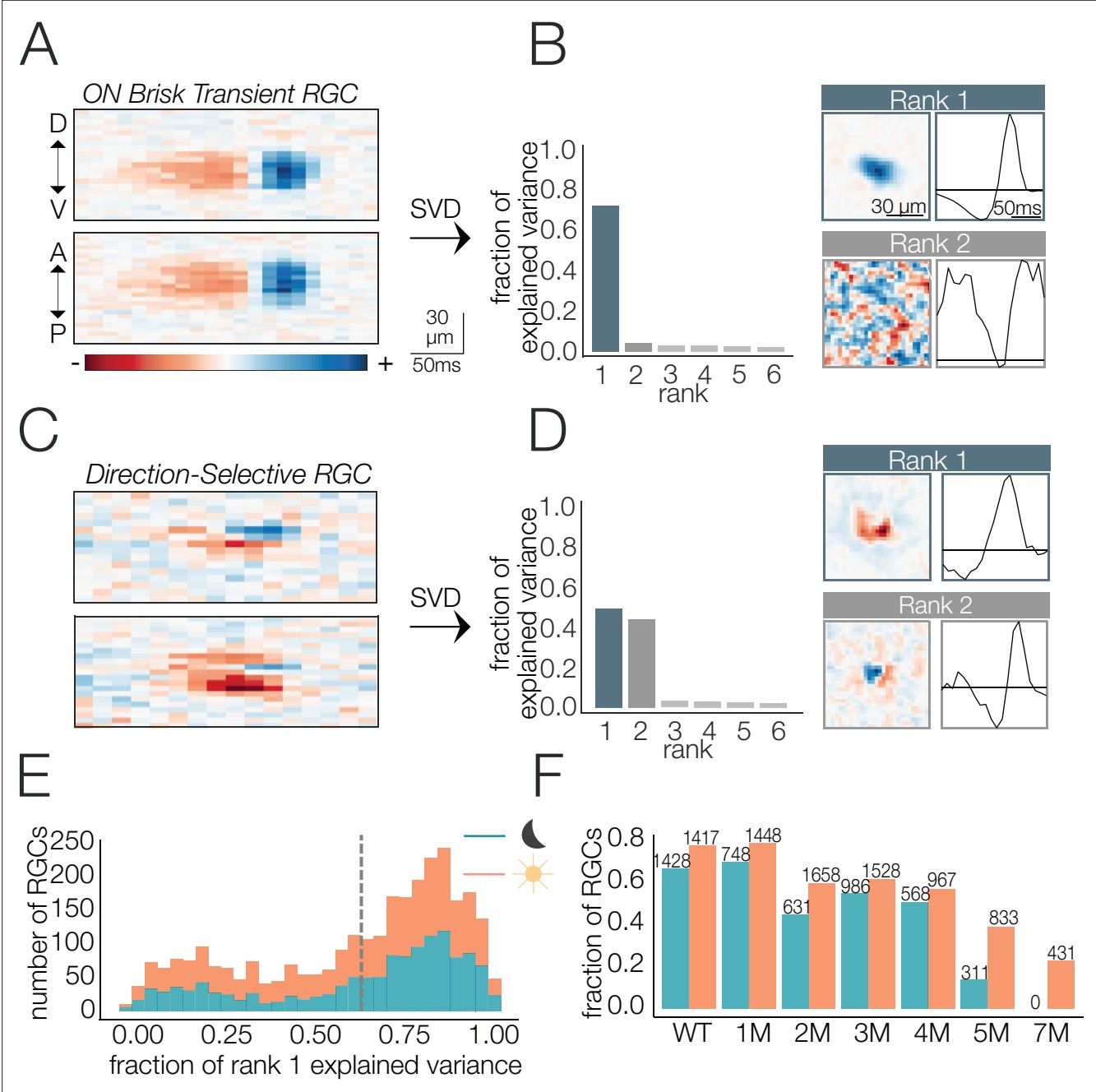

**Figure 2.** Identifying spatial and temporal receptive field (RF) components. (**A**) Example retinal ganglion cell (RGC) spike-triggered average stimulus (STA) that is space-time-separable. Top and bottom are space-time plots along orthogonal spatial dimensions: D, V, A, and P indicate dorsal, ventral, anterior, and posterior directions. (**B**) Singular value decomposition (SVD) of STA in (**A**); (left) distribution of variance explained by first six space-time vector pairs; (top right: rank 1) spatial (left) and temporal (right) filters from the rank 1 decomposition; (bottom right; rank 2) spatial and temporal filters associated with the second singular value. (**C**) Example RGC STA that was not space-time-separable. (**D**) SVD of STA in (**C**); (left) distribution of variance explained by first six space-time vector pairs; (top right; rank 1) spatial and temporal filters from the rank 1 decomposition; (bottom right; rank 2) spatial and temporal filters associated with the second singular value. (**E**) Distribution of variance explained by the rank 1 decomposition for all wild-type (WT) RGCs (1654 cells from five mice) under mesopic (teal) and photopic (orange) conditions. Vertical line shows threshold for classifying cells as space-time-separable. (**F**) Fraction and quantity of RGCs measured at mesopic (teal) and photopic (orange) light levels that were classified as exhibiting space-time-separable RFs. Source files for (**E**) and (**F**) are available in *Figure 2—source data 1*.

The online version of this article includes the following source data and figure supplement(s) for figure 2:

**Source data 1.** Contains counts of RGCs with space-time separable RFs.

*Figure 2 continued on next page*

*Figure 2 continued*

**Figure supplement 1.** Linear–nonlinear (LN) model performance and fraction of light-responsive retinal ganglion cells (RGCs) as a function of rod death.

space-time-separable STA will yield one pair of spatial and temporal filters that captures most of the variance in the STA (a.k.a., rank 1 approximation, *Figure 2A and B*). Subsequent space-time filter pairs will exhibit little to no structure and appear as 'noise' (*Figure 2B*, bottom right). In contrast, nonseparable STAs will require multiple space-time vector pairs to reproduce the STA (*Figure 2C and D*) and each pair will exhibit clear structure in space and time (*Figure 2D*). RGCs for which >60% of the variance in the full space-time STA was captured by a separable RF model were included in the analysis. This threshold was chosen because it captured a clear mode in the distribution of all RGCs from WT retinas (*Figure 2E*). Across timepoints of degeneration, 8028 out of 12,997 RGCs met this criterion. However, the fraction passing this criterion changed as a function of degeneration, with more cells passing the criterion early in degeneration and fewer passing at the latest timepoints (*Figure 2F*). RGCs passing this criterion had a wide range of spatial RF sizes and temporal RF durations and consisted of both ON and OFF classes. Thus, several, but not all, RGCs met this criterion. The decrease in the fraction of RGCs with space-time-separable RFs at the latest stages of degeneration (5–7 months; *Figure 2F*) and under photopic conditions was accompanied by a decrease in the total fraction of RGCs that were light responsive (*Figure 2—figure supplement 1D*). 'Light responsive' was defined by how strongly the spike rate was modulated by the checkerboard stimulus (see 'Materials and methods'). Under mesopic conditions, this decrease in the fraction of responsive RGCs occurred earlier (2 months), probably because of rapidly deteriorating rods (*Figure 2—figure supplement 1C*). Below we analyze changes in the spatial and temporal RFs of RGCs at mesopic and photopic light levels as a function of photoreceptor degeneration.

## Changes in mesopic receptive fields with photoreceptor degeneration

At the mesopic light level, the distribution of spatial RF sizes between WT and 1-month *Cngb1^neo/neo^* animals was relatively stable (*Figure 3A and B*). A small but statistically significant decrease in the mean RF sizes relative to WT was observed at 1–4 months (15% difference between WT to 4 months; p-value: 0.039). At 5 months, the decrease was attenuated. By 7 months, no RGCs exhibited space-time-separable RFs at the mesopic light level (*Figure 2F*) and only 30% of identified RGCs exhibited visual responses (*Figure 2—figure supplement 1C*). Note that corrupting a rank 1 STA with increasing amounts of noise will cause the rank 1 approximation to capture less total variance, which likely accounts for the reduced number of RFs at the latest timepoints.

At the mesopic condition, we observed larger fractional changes in RGC temporal integration than in spatial integration between WT and *Cngb1^neo/neo^* mice (*Figure 3C and D*). At 1–3 months, the mean temporal integration and variance were larger for *Cngb1^neo/neo^* animals than WT: the mean of all RGCs increased by 14, 17, and 13% at 1, 2, and 3 months, respectively (p-value: 0.10, 0.08, 0.11), and the variance of all RGCs increased by 38, 34, and 35% at 1, 2, and 3 months of age respectively (p-value: 0.01, 0.02, 0.02). At 4 months, the median temporal integration increased further by 88% relative to WT (p-value: 0.001). However, at 5 months, the temporal integration decreased back toward WT (9% increase in time-to-zero relative to WT; p-value: 0.14), with the variance in the distribution substantially larger (37% increase in variance relative to WT; p-value: 0.009). Overall, spatial RF structure was relatively stable until the latest stages of degeneration: the largest change from WT in spatial integration was 15% (at 4 months). Changes in temporal integration were greater, with the largest change being 88% (4 months), but these changes fluctuated over time and were non-monotonic with degeneration (see 'Discussion').

We also analyzed the spatial and temporal RF properties of a specific RGC type, ON brisk-sustained RGCs, which exhibited a mosaic-like pattern of RFs across the MEA (*Figure 2—figure supplement 1*; *Ravi et al., 2018*). These RGCs have space-time-separable RFs and could be readily identified by their spike train autocorrelation functions. This irreducible RGC type tracked the spatial and temporal RF trends displayed in the entire population (dark gray distribution, *Figure 3*), suggesting that trends across the population were representative of trends in individual cell types. To check that these trends were not driven by experiment-to-experiment variability, we also analyzed changes in RF structure by

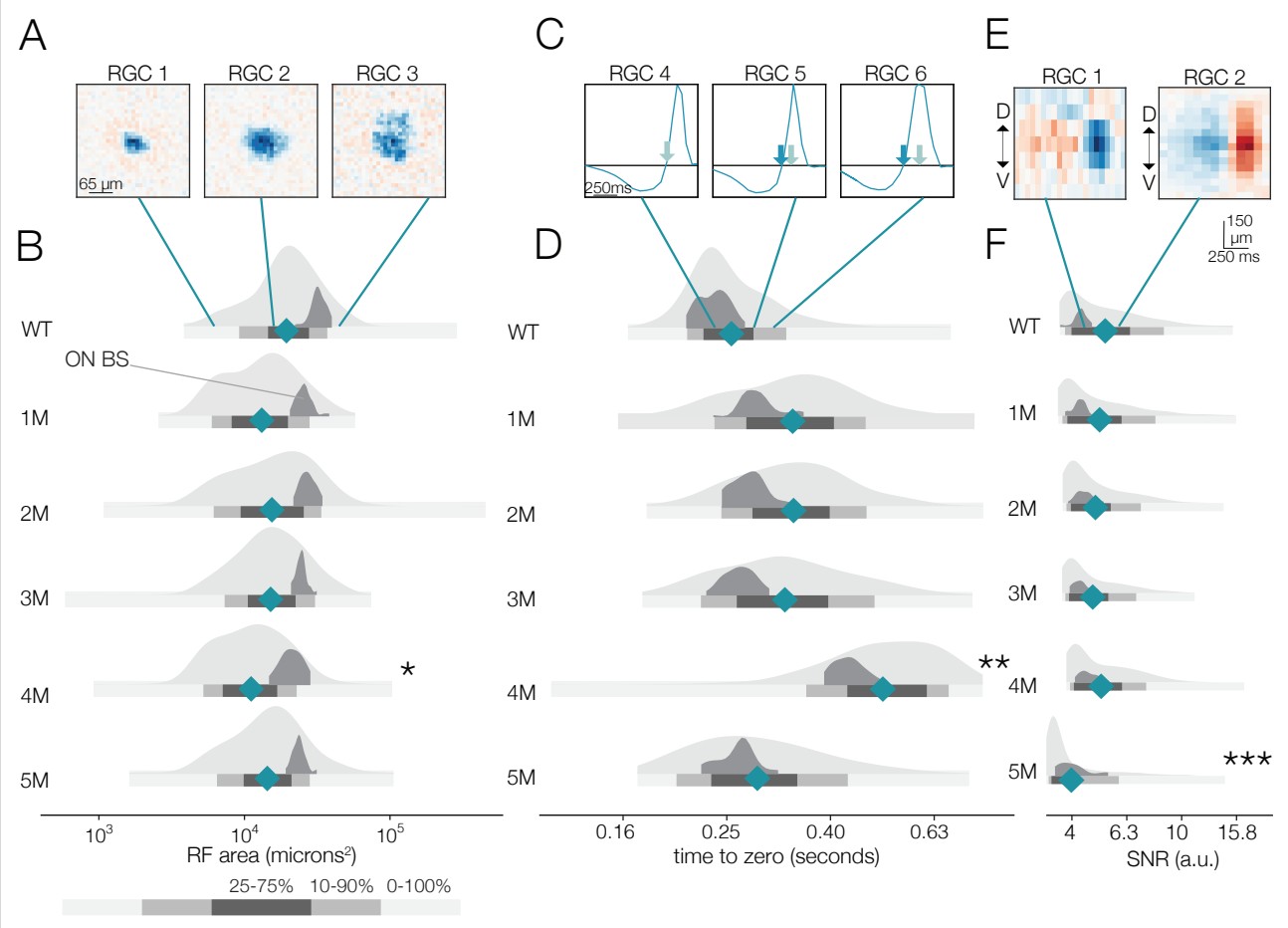

**Figure 3.** Changes in receptive field (RF) structure under mesopic conditions induced by rod death. (**A**) Example spatial RFs at mesopic light levels showing smaller to larger RFs (left to right). (**B**) Distributions of spatial RF center areas comparing wild-type (WT) and *Cngb1neo/neo* mice from 1 to 5 months of degeneration for all retinal ganglion cells (RGCs) with space-time-separable RFs (light gray) and ON brisk-sustained RGCs (dark gray). Blue diamonds indicate the mean of each light gray distribution. (**C**) Example temporal RFs showing briefer to longer integration. Light blue arrows indicate the time to zero for RGC 4; dark blue arrows show time to zero for RGCs 5 and 6. (**D**) Same as (**B**) but showing distributions of the temporal RF times-to-peak. (**E**) Example spike-triggered average stimulus (STAs) with lower (left) and higher (right) signal-to-noise ratios (SNRs). (**F**) Same as (**B**) but showing distribution of STA SNRs. Grayscale legend indicates percentile ranges of distributions in (**B**), (**D**), and (**F**) for comparing dispersion across distributions. Stars indicate significant changes in the mean from WT for population data (light gray distributions): *, **, and *** are p<0.1, 0.01, and 0.001, respectively. Source files for (**B**), (**D**), and (**F**) are available in *Figure 3—source data 1*.

The online version of this article includes the following source data and figure supplement(s) for figure 3:

**Source data 1.** Contains values for distributions shown in panels B, D, and F.

**Figure supplement 1.** Trends in mesopic receptive field (RF) structure during rod death are consistent across experiments.

experiment, rather than pooling all RGCs at individual timepoints (*Figure 3—figure supplement 1*). These trends were not driven by experimental variance.

Given that photoreceptors are dying, it is potentially surprising that spatiotemporal RF structure is relatively stable under mesopic conditions. Fewer photoreceptors ought to result in diminished sensitivity, even if the area of spatial integration or the duration of temporal integration is relatively stable. Thus, we analyzed the signal-to-noise ratio (SNR) of the STAs that may be expected to be noisier with photoreceptor death (see 'Materials and methods'). The SNR of the STAs generally drifted down over time (*Figure 3E and F*); however, even these changes were relatively small until 5 months (compared to WT, there was a 5 and 15% decline in SNR in the 4- and 5-month experimental groups, respectively; p-value: 0.19 and 0.04). This trend was preserved in the ON brisk-sustained population (*Figure 3F*, dark gray distributions). Of note, the STA measurements were based on 30 min of checkerboard stimuli, which may be a sufficiently long period of time to average away noise and/or compensate

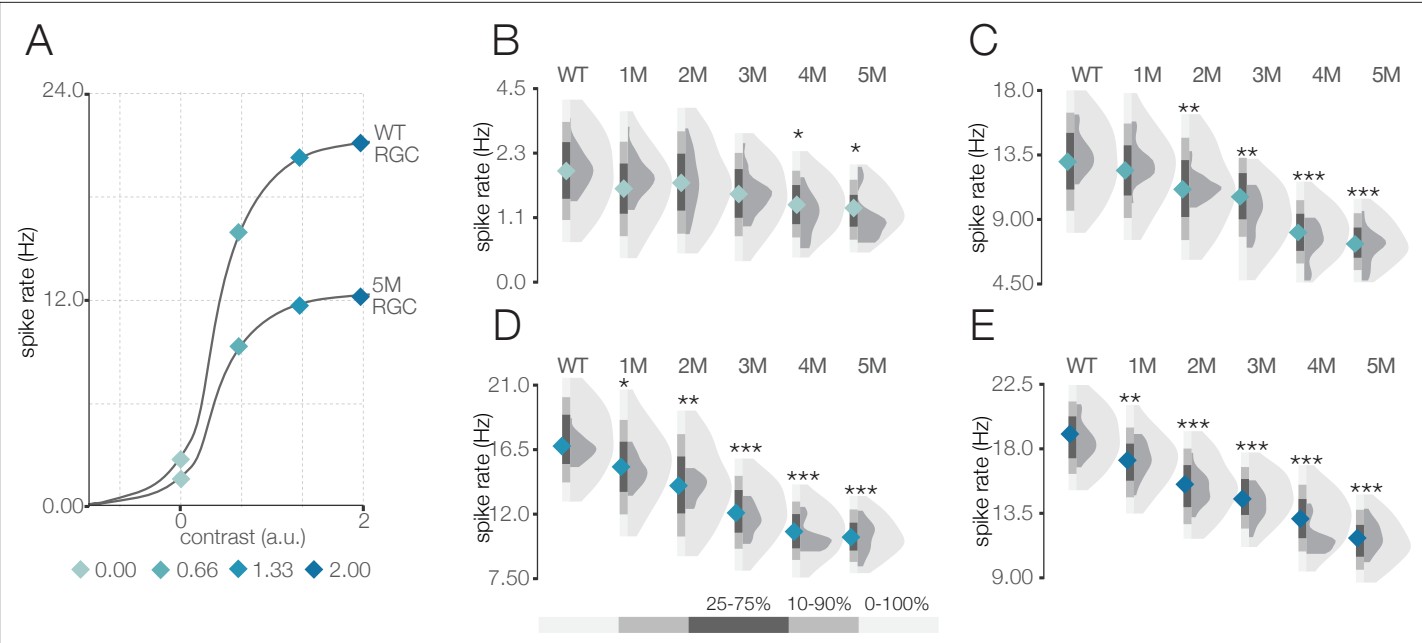

**Figure 4.** Response gain steadily decreases during rod death under mesopic conditions. (**A**) Mean cumulative Gaussian fit to contrast response functions across retinal ganglion cells (RGCs) from wild-type (WT) and 5-month *Cngb1neo/neo* mice. Four locations along the contrast response functions are highlighted by blue diamonds. (**B–E**) Distributions of contrast response function values at contrasts indicated by blue diamonds in (**A**) for WT and *Cngb1neo/neo* 1- to 5-month retinas. Light and dark gray distribution are all RGCs and ON brisk-sustained RGCs, respectively. Diamond at the base of each distribution indicates mean of all RGCs (light gray distribution). Grayscale legend indicates percentile ranges of distributions for comparing dispersion across distributions. Source files for (**B–E**) are available in *Figure 4—source data 1*.

The online version of this article includes the following source data and figure supplement(s) for figure 4:

**Source data 1.** Contains values for distributions in panels B, C, D and E.

**Figure supplement 1.** Trends in mesopic contrast response functions during rod death are consistent across experiments.

for diminished gain. To more directly assess changes in response gain, we inspected the contrast response functions (also called 'static nonlinearities') across RGCs, which capture the relationship between the stimulus (filtered by the RF) and the spiking output (*Chichilnisky, 2001*). The gain of these contrast response functions steadily decreased as a function of degeneration (*Figure 4*; also observed for ON brisk-sustained cells shown in dark gray distributions). This trend was not driven by experiment-to-experiment variability (*Figure 4—figure supplement 1*). The effect was most noticeable for stimuli that were most similar to the RF (*Figure 4C–E*). Thus, under mesopic conditions, gain steadily decreased with photoreceptor degeneration, while spatial and temporal integration was relatively stable until the latest stages of degeneration.

### Changes in photopic receptive fields with photoreceptor degeneration

Under photopic conditions, the distribution of spatial RF sizes was remarkably stable throughout degeneration (*Figure 5A and B*). The largest change was at 7 months with a 5% decrease in spatial RF area relative to WT (p-value: 0.211). Note that, unlike the mesopic condition, space-time-separable RFs were detectable out to 7 months under the photopic condition; however, there was a decrease in the number of light-responsive RGCs beginning at 5 months (*Figure 2—figure supplement 1D*).

Temporal RFs exhibited greater changes than spatial RFs under photopic conditions (*Figure 5C and D*) but were still surprisingly stable. Specifically, the time to zero progressively decreased 10, 12, and 15% from 1 to 3 months, respectively (relative to WT; p-value: 0.32, 0.22, 0.04), indicating a shortening of temporal integration. At 4 months, the temporal integration slowed somewhat, becoming 18% slower than 3 months but only 4% slower than WT (p-value: 0.423) (*Figure 5D*). At 5 and 7 months, time to zero decreased by 12 and 13% (p-value: 0.22, 0.12), respectively, relative to 4 months.

These changes in temporal and spatial RF structure under photopic conditions were relatively small given the clear changes in cone morphology observed, particularly at 5–7 months (*Figure 1*): the

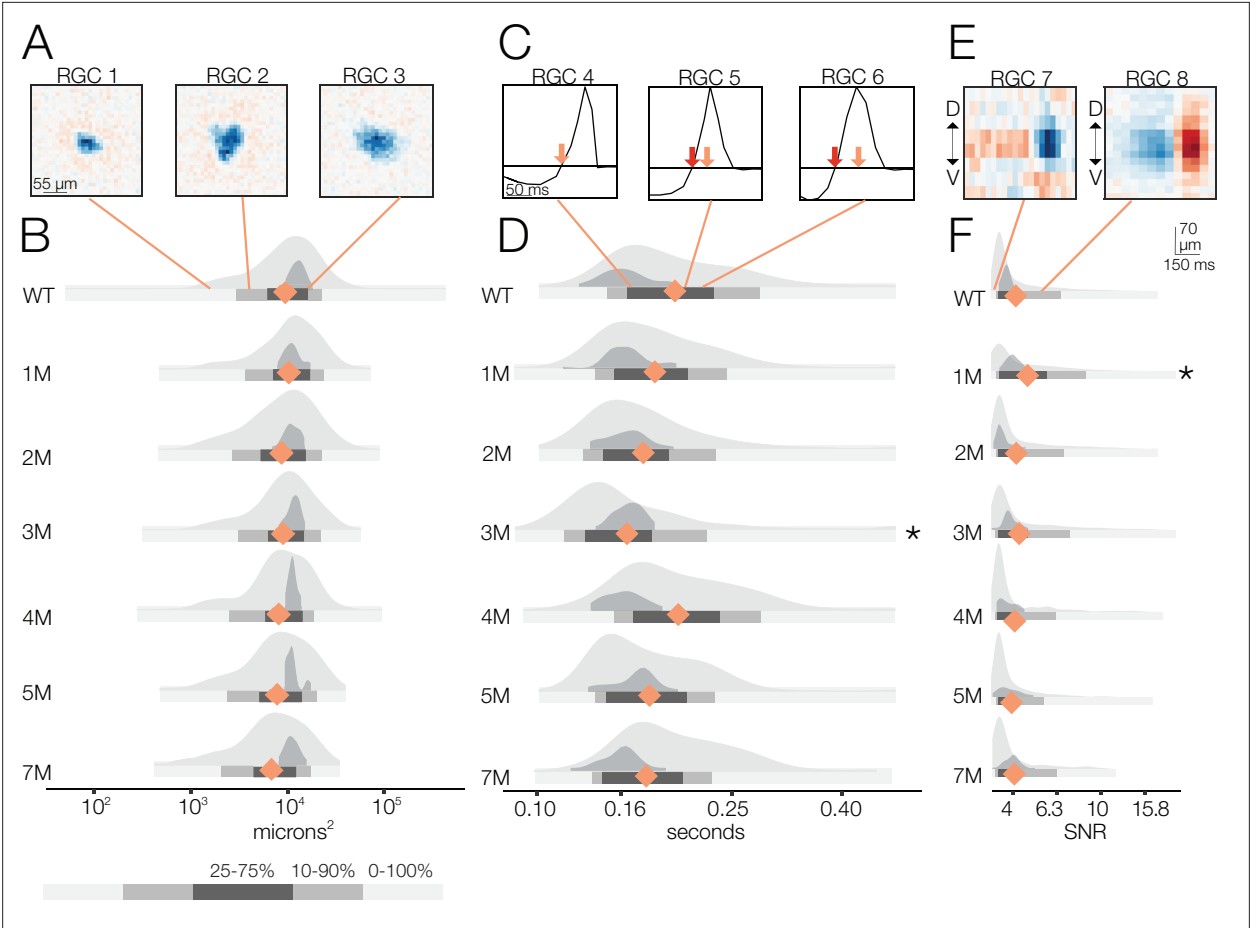

**Figure 5.** Changes in receptive field (RF) structure under photopic conditions induced by rod death. (**A**) Example spatial RFs at the photopic light level showing small to large RF areas (left to right). (**B**) Distributions of spatial RFs comparing wild-type (WT) and *Cngb1neo/neo* retinas from 1 to 7 months of degeneration for all retinal ganglion cells (RGCs) with space-time-separable RFs (light gray) and ON brisk-sustained (dark gray). Orange diamonds indicate the mean of each light gray distribution. (**C**) Example temporal RFs at the photopic light level. Orange arrows indicate the time to zero for RGC 4; red arrows show time to zero for RGCs 5 and 6. (**D**) Distributions of temporal RFs comparing WT and *Cngb1neo/neo* retinas from 1 to 7 months of degeneration. (**E**) Example spike-triggered average stimulus (STAs) with lower and higher signal-to-noise ratios (SNRs). (**F**) Distribution of SNRs comparing WT and *Cngb1neo/neo* mice. Grayscale legend indicates percentile ranges of distributions in (**B**), (**D**), and (**F**) for comparing dispersion across distributions. Source files for (**B**), (**D**), and (**F**) are available in *Figure 5—source data 1*.

The online version of this article includes the following source data and figure supplement(s) for figure 5:

**Source data 1.** Contains values for distributions shown in panels B, D, and F.

**Figure supplement 1.** Trends in photopic receptive field (RF) structure during rod death are consistent across experiments.

largest deviation in mean temporal integration from WT occurred at 3 months, and this was only a 15% slowing in the time-to-zero, while spatial RF sizes changed by 5% at most (relative to WT) through 7 months.

We next analyzed the SNR of the STAs (*Figure 5E and F*). We observed a decline in SNR with degeneration under photopic conditions. Yet even at 7 months, with 30% cone loss, near complete rod loss, and abnormal cone morphologies, there was only a 6% decrease in the median SNR (*Figure 5F*; p-value: 0.456). As with the mesopic conditions, computing STAs over relatively long periods of checkerboard noise may mask decreases in gain or increases in response noise. To inspect changes in gain, we analyzed the contrast response functions (*Figure 6*). Like the mesopic condition, response gain steadily decreased with degeneration, particularly for stimuli that increasingly matched the RF (*Figure 6D and E*).

These RGC population trends in spatial and temporal RF changes and contrast response functions over time were mirrored by the ON brisk-sustained RGCs (dark gray distributions in *Figure 5B, D, and*

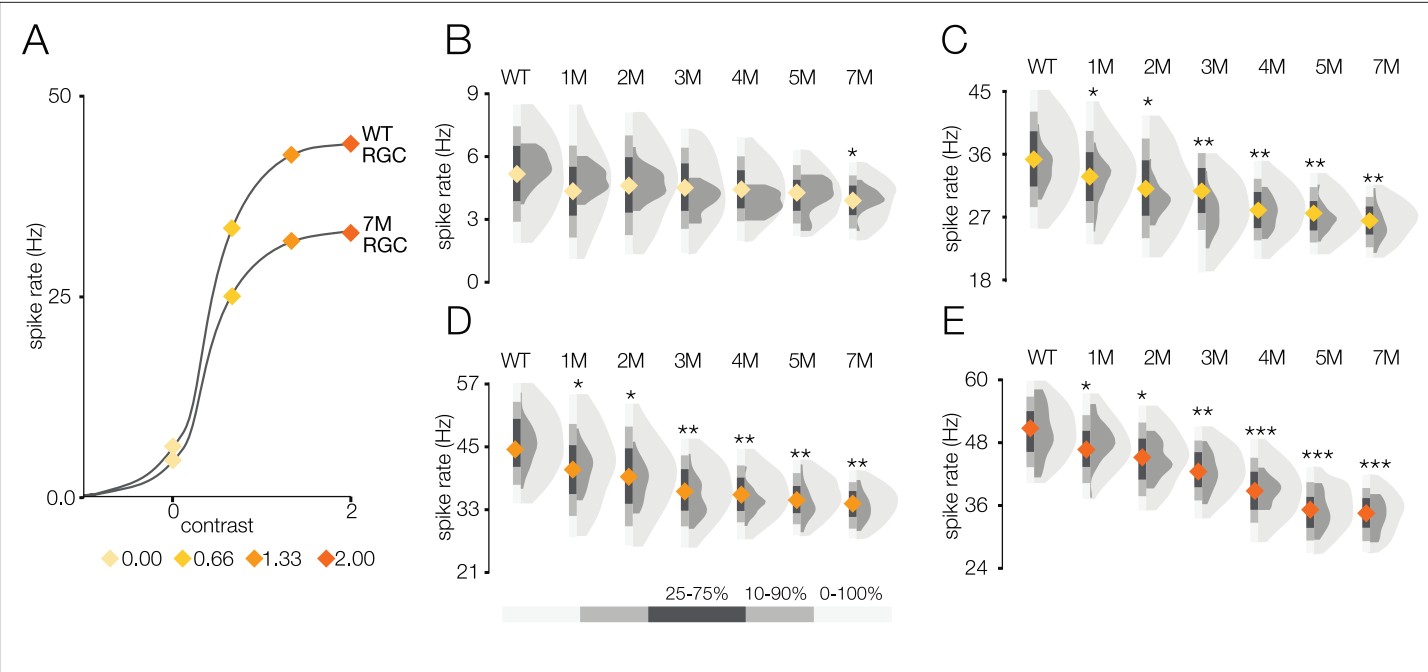

**Figure 6.** Response gain steadily decreases during rod death under photopic conditions. (**A**) Mean cumulative Gaussian fit to contrast response functions across retinal ganglion cells (RGCs) from wild-type (WT) and 7-month *Cngb1neo/neo* mice. Four locations along the contrast response functions are highlighted by diamonds. (**B–E**) Distribution of contrast response function values at contrast values indicated in (**A**) for WT and 1- to 7-month retinas. Light and dark gray distribution are all RGCs and ON brisk-sustained RGCs, respectively. Diamond at the base of each distribution indicates mean of all RGCs with space-time-separable receptive fields (RFs) (light gray distribution). Grayscale legend indicates percentile ranges of distributions for comparing dispersion across distributions. Source files for (**B–E**) are available in *Figure 6—source data 1*.

The online version of this article includes the following source data and figure supplement(s) for figure 6:

**Source data 1.** Contains values for distributions shown in panels B, C, D, and E.

**Figure supplement 1.** Trends in photopic contrast response functions during rod death are consistent across experiments.

*F*, *Figure 6B and C*), suggesting that they were general across cell types. Furthermore, these trends were not driven by experiment-to-experiment variability (*Figure 5—figure supplement 1*, *Figure 6—figure supplement 1*). In summary, the dominant effect of photoreceptor degeneration on photopic RF structure among RGCs in *Cngb1neo/neo* mice was a decrease in response gain, not a change in spatial or temporal integration.

### *Cngb1neo/neo* RGCs do not exhibit oscillatory activity while photoreceptors remain

The results above indicate relatively small changes in spatial and temporal integration of cone-mediated responses as rods die and cones degenerate in *Cngb1neo/neo* mice. They also indicate clear decreases in gain as a function of degeneration under mesopic and photopic conditions (*Figures 4 and 6*). However, the analyses above do not provide much insight into the extent that noise may be changing with degeneration. One source of noise is signal-independent increases in spontaneous activity. Several rodent models of retinal degeneration exhibit increased spontaneous activity arising as 5–10 Hz oscillations in RGC spiking (*Marc et al., 2007*; *Margolis et al., 2008*; *Stasheff, 2008*). These oscillations are reported to occur throughout the degeneration process (*Stasheff et al., 2011*). The presence of such oscillations may minimally impact the RF measurements as they would be largely averaged away when computing the STA, but they could contribute to the lower SNR of the STAs (*Figures 3F and 5F*) along with the decreased gain (*Figures 4 and 6*).

Thus, we analyzed the spontaneous activity of RGCs in WT and *Cngb1neo/neo* animals from 1 to 9 months to check for oscillatory activity (*Figure 7*). Spontaneous activity was measured in darkness as well as with static illumination set to the same mean intensity as the checkerboard stimuli presented at the mesopic and photopic light levels. Oscillations were absent in the spiking activity of

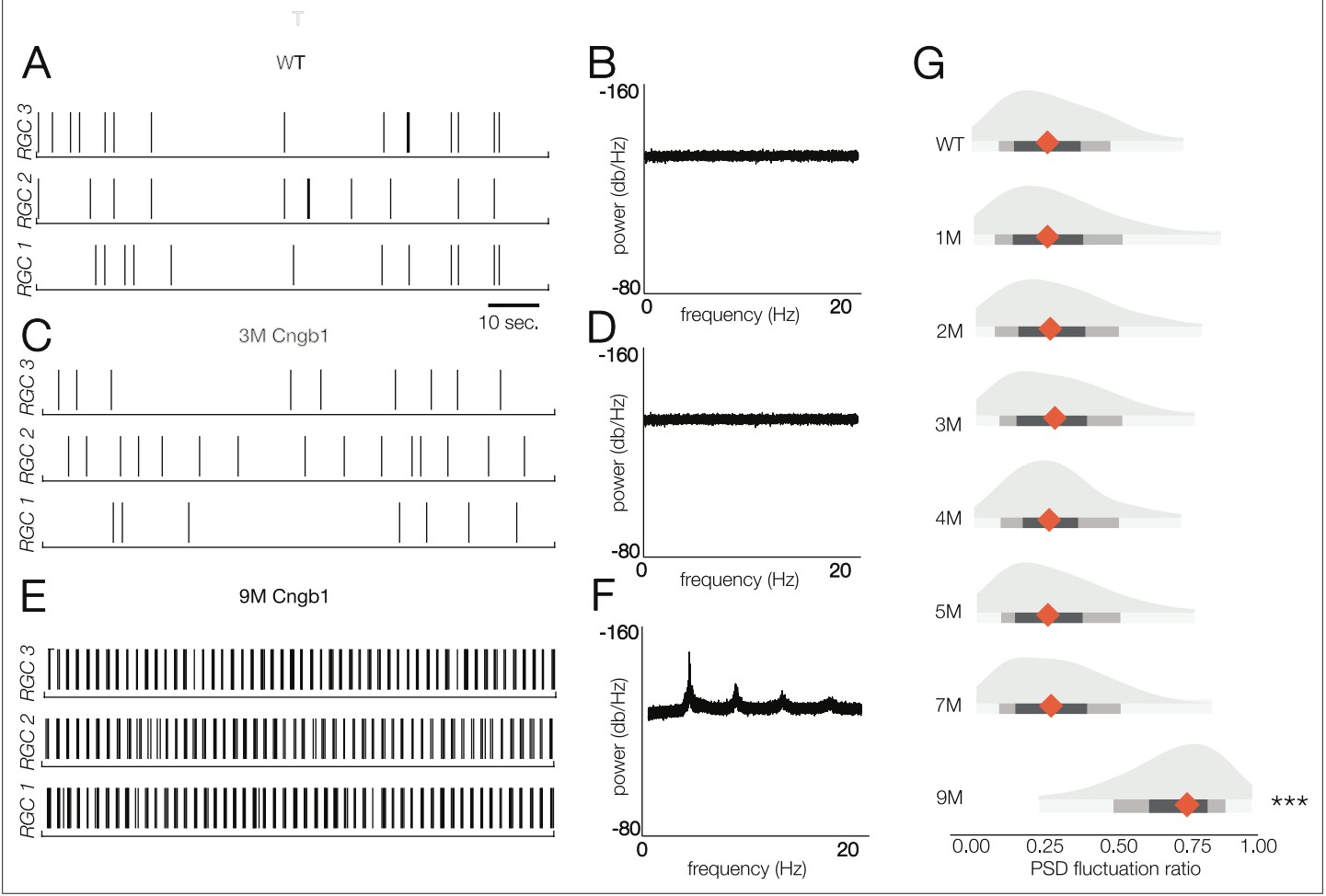

**Figure 7.** Retinal oscillations occur after photoreceptor loss in *Cngb1^neo/neo* retinas. (**A**) Spontaneous activity of three representative wild-type (WT) retinal ganglion cells (RGCs) in total darkness. (**B**) The power spectral density (PSD) of one example RGC. (**C, D**) Example rasters (**C**) and PSD (**D**) of a representative 3-month *Cngb1^neo/neo* RGC. (**E, F**) Example rasters and the PSD of a representative 9-month *Cngb1^neo/neo* RGC. (**G**) Distributions of the PSD fluctuation ratios, defined as the maximum of PSD divided by the baseline PSD (see 'Materials and methods'). Mean indicated by orange diamond, gray bars indicate 100% (light), 80% (medium), and 50% (dark) percentile ranges. Source files for (**G**) are available in *Figure 7—source data 1*.

The online version of this article includes the following source data for figure 7:

**Source data 1.** Contains values for distributions shown in panel G.

all recorded RGCs in darkness and at both mesopic and photopic light levels in *Cngb1^neo/neo* retinas from 1 to 7 months (*Figure 7A–D and G*). However, clear 5 Hz oscillatory activity did arise at 9 months (*Figure 7E–G*). The oscillations were present at all three light levels (not shown). At 9 months, no light response could be detected with 100% contrast flashes at the mesopic or photopic light levels (not shown, n = 2). Additionally, while a small number of cells labeled for cone arrestin at 9 months, M-opsin expression was barely, if at all, detectable (*Figure 1—figure supplement 1*). Thus, oscillations in the spontaneous spiking activity of RGCs in the *Cngb1^neo/neo* mouse model are absent until all (or nearly all) of the RGCs have lost their light responses (see 'Discussion').

## *Cngb1^neo/neo* RGCs exhibit deteriorated visual signaling under mesopic conditions

Thus far, we have observed a decrease in response gain among RGCs under mesopic and photopic conditions that tracks progressive photoreceptor degeneration. Furthermore, signal-independent noise, as assayed by changes in spontaneous activity, appears constant until all (or nearly all) photoreceptors have died in *Cngb1^neo/neo* mice. It is possible that the RGCs are signaling less reliably as photoreceptors die: the STA method averages over many trials and is thus relatively insensitive to potential

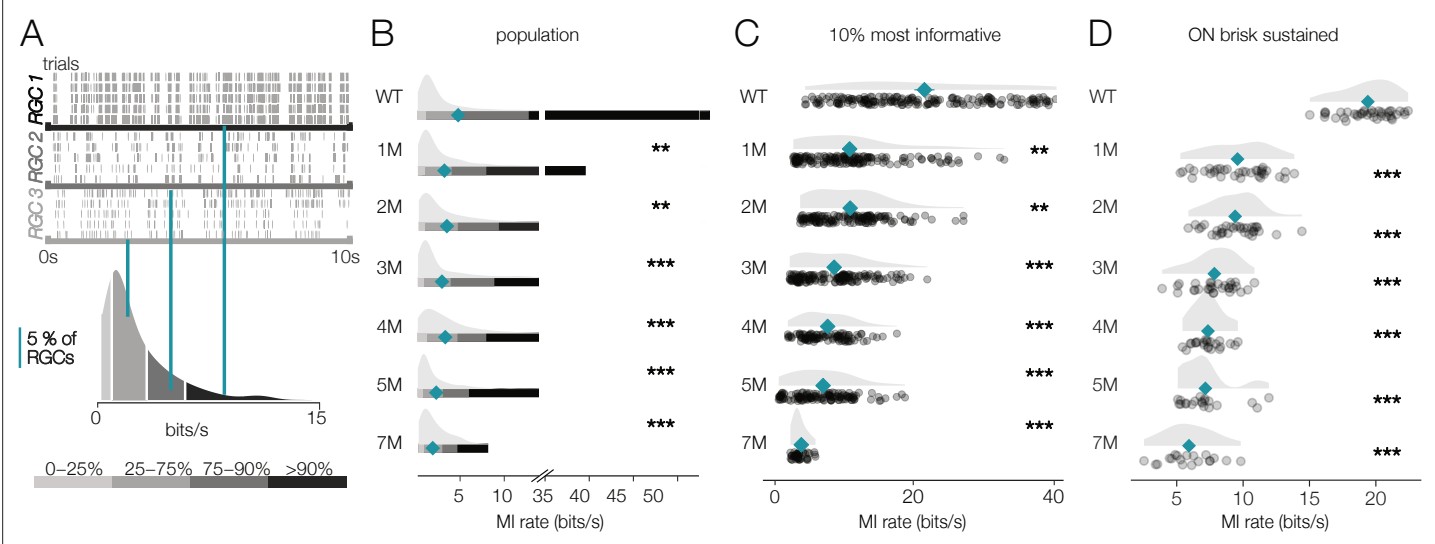

**Figure 8.** Retinal ganglion cell (RGC) signaling fidelity at mesopic condition decreases with rod death. (**A**) (Top) Rasters from three example RGCs responding to a repeated mesopic checkerboard stimulus. (Bottom) Distribution of information rates across all RGCs from one experiment; teal lines show where each example RGC lies in the distribution. (**B**) Distributions of information rates for all RGCs in each condition: wild-type (WT) and *Cngb1^neo/neo* retinas. Mean shown by teal diamonds, stars indicate significant changes from WT: *, **, and *** are p<0.1, 0.01, and 0.001, respectively. (**C**) Distributions of information rates of 10% most informative RGCs across conditions. (**D**) Distributions of information rates for ON brisk-sustained RGCs across conditions. Source files for (**B–D**) are available in *Figure 8—source data 1*.

The online version of this article includes the following source data and figure supplement(s) for figure 8:

**Source data 1.** Contains values for distributions shown in panels B, C, and D.

**Figure supplement 1.** Mesopic information rate changes are not driven by experiment-to-experiment variability.

changes in signal-dependent noise that could manifest as increased variability in either the number or timing of spikes produced by a stimulus. Because of this, it is important to identify potential changes in the fidelity of RGC signaling that may accompany photoreceptor degeneration.

To measure the fidelity of RGC signaling, we used information theory (*Shannon, 1948*). Specifically, we calculated the mutual information between spike trains and the stimulus (see 'Materials and methods'). Mutual information indicates how much the uncertainty about the visual stimulus is reduced by observing the spike train of an RGC (see review by *McDonnell et al., 2011*). In general, if the RGC response is highly variable, observing a single response will reduce uncertainty about the stimulus less than for an RGC with highly reproducible responses.

We began by presenting a repeating checkerboard stimulus at the mesopic light level (~100 Rh*/rod/s). We segregated RGCs based on their information rate for the checkerboard stimulus (*Figure 8*). Some RGCs exhibited strongly modulated and reproducible responses across repetitions of the stimulus (*Figure 8A*, RGC 1), while other RGCs were less reliably driven by the stimulus (*Figure 8A*, RGCs 2 and 3). Note that all RGCs with a spike rate above 3 Hz (12,997 RGCs) were used in this analysis. This analysis included RGCs with space-time-separable RFs that were well-fit by the linear–nonlinear model used in *Figures 2–6*, and cells without space-time-separable RFs that were not well described by an LN model.

For WT retinas, the distribution of information rates across RGCs was approximately unimodal with a long tail skewed toward high rates (*Figure 8A*, bottom). The peak of these distributions did not change much with degeneration (*Figure 8B*). However, the peak primarily represented cells that were not particularly informative about the checkerboard stimulus even in WT retinas. In contrast, the median of the information rate distributions decreased as photoreceptors died (*Figure 8B*, teal diamonds). This change in the median was largely driven by decreased information rates in the long tails of these distributions (*Figure 8B*). We analyzed the 10% of RGCs that were most informative about the checkerboard stimulus: those with the highest information rates (*Figure 8C*). For these RGCs, there was a clear drop in information rates between WT and 1-month *Cngb1^neo/neo* mice (mean of all RGCs changed from 5.7 bits/s to 4.3 bits/s; p-value: 0.003). In fact, *Cngb1^neo/neo* RGCs at all

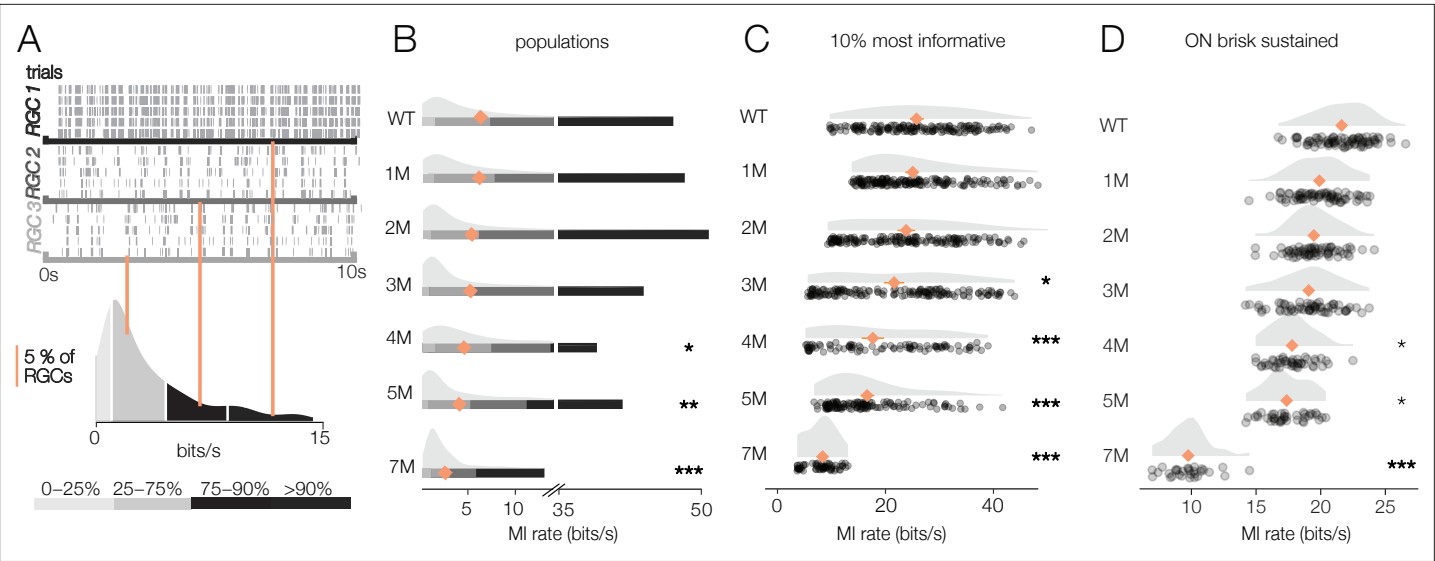

**Figure 9.** Retinal ganglion cell (RGC) signaling fidelity at photopic condition is relatively stable with rod death. (**A**) (Top) Rasters from three example RGC responding to a repeated photopic checkerboard stimulus. (Bottom) Distribution of information rates across all RGCs from one experiment; orange lines show where each example RGC lies in the distribution. (**B**) Distributions of information rates of for all RGCs in each condition: wild-type (WT) and *Cngb1^neo/neo^* retinas. Means shown by orange diamonds. (**C**) Distribution of information rates of 10% most informative RGCs across conditions. (**D**) Distribution of information rates for ON brisk-sustained RGCs across conditions. Source files for (**B**–**D**) are available in *Figure 9—source data 1*.

The online version of this article includes the following source data and figure supplement(s) for figure 9:

**Source data 1.** Contains values for distributions shown in panels B, C, and D.

**Figure supplement 1.** Photopic information rate changes are not driven by experiment-to-experiment variability.

timepoints examined transmitted significantly less information than WT RGCs at the mesopic light level (mean of all RGCs ranged from 2.1 to 4.3 bits/s). Information rates between 1 and 2 months did not change significantly (difference of 0.09 bits/s was observed; p-value: 0.655), but there was a steady decline at subsequent timepoints. ON brisk-sustained RGCs exhibited similar information rate trends: a large drop in information between WT and 1 month, with a steady decrease as degeneration progressed (*Figure 8D*). The trends in the entire population and for ON brisk-sustained RGCs were not driven by experiment-to-experiment variability (*Figure 8—figure supplement 1*).

These data indicate that under low mesopic conditions there is a relatively steady decline in the reliability with which RGCs signal visual information in *Cngb1^neo/neo^* mice. The difference between WT and 1- to 2-month *Cngb1^neo/neo^* animals is likely because rods are poorly responsive to light in *Cngb1^neo/neo^* animals due to a greatly reduced photocurrent (*Wang et al., 2019*). The progressive decline in information rates after 2 months is likely caused by large-scale loss of rods (*Figure 1A*), and possibly a reduced ability of cones to signal near their threshold for activation.

### *Cngb1^neo/neo^* RGCs exhibit relatively stable visual signaling at photopic conditions

We next checked the reliability of RGC signaling under photopic conditions that are well above the threshold of cone vision (10,000 R*/rod/s). It is possible that at this higher light level visual signaling is more (or less) stable. Similar to the mesopic level, RGCs exhibited a wide range of information rates to checkerboard stimuli, with some exhibiting very reliable responses to a repeated stimulus while others exhibited more variable responses (*Figure 9A*). Information rates were much more stable among RGCs at these higher light levels, with no significant differences in median information rates observed between WT and 1–3 months of degeneration (p-values of 0.56, 0.32, 0.29 between WT and 1, 2, and 3 months, respectively) (*Figure 9B*). Even when focusing on the top 10% of most informative RGCs, 1 and 2 months of degeneration were not significantly different from WT (*Figure 9C*). When focusing on a single-cell type, the ON brisk-sustained RGCs also exhibited quite stable information rates (*Figure 9D*). Despite clear changes in cone morphology at 5 months (*Figure 1*), decreases in

information rates were modest (mean of top 10% of RGCs changed from 23.1 bits/s in WT to 18.1 bits/s at 5 months; p-value: 0.0009; mean of ON brisk-sustained population changed from 21.5 bits/s to 17.6 bits/s). At 7 months, RGCs continued to transmit cone-mediated information, but with a relatively pronounced decrease (mean rate was 8.6 bits/s among the top 10% of RGCs and 10.1 bits/s among ON brisk-sustained RGCs). Overall, these data indicate that the fidelity of cone-mediated visual signaling among RGCs is quite stable until ~2–3 months of degeneration, corresponding to 50–70% rod loss.

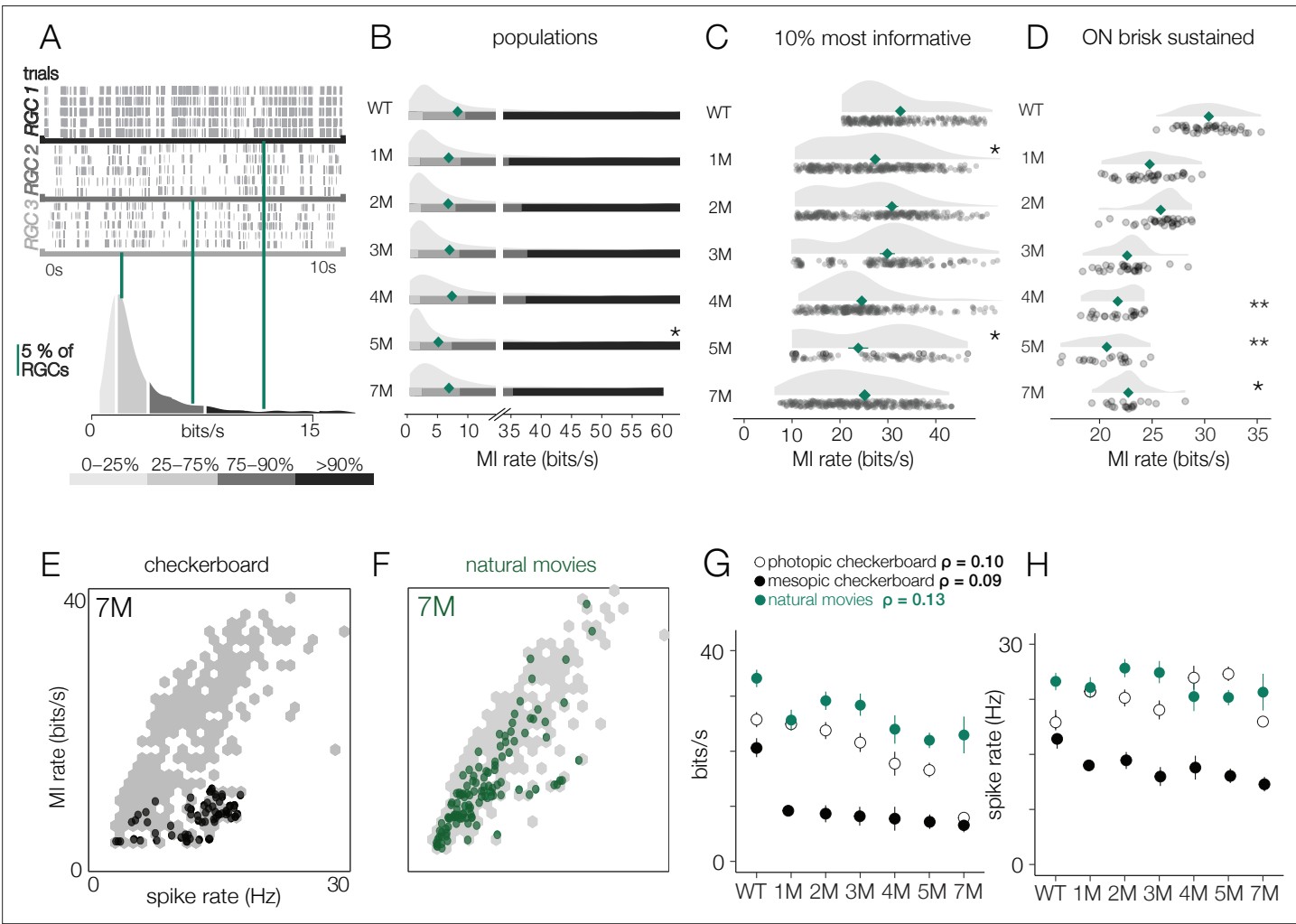

**Figure 10.** Retinal ganglion cell (RGC) signal fidelity is higher for natural movies than checkerboard noise late into degeneration. (**A**) (Top) Rasters from three example RGCs responding to a repeated photopic natural movie stimulus. (Bottom) Distribution of information rates across all RGCs from one experiment; green lines show where each example RGC lies in the distribution. (**B**) Distributions of information rates to a natural movie for all RGCs in each condition for wild-type (WT) and *Cngb1^neo/neo* retinas. Mean shown by green diamonds. (**C**) Distribution of information rates to a natural movie for the 10% most informative RGCs across conditions. (**D**) Distribution of information rates to a natural movie for ON brisk-sustained RGCs across conditions. (**E, F**) Scatter plot of information rates and spike rates from RGC responses to photopic checkerboard stimulus (**E**) or a natural movie (**F**). Gray dots are RGC responses from the total population (all conditions). Black (**E**) and green (**F**) dots are from 7-month *Cngb1^neo/neo*. (**G**) Mean information rates of RGC responses to natural movies (green), photopic checkerboard (open), and mesopic checkerboard (black) across WT and *Cngb1^neo/neo* retinas. Error bars are 2× SE. (**H**) Mean spike rates of RGC responses to natural movies (green), photopic checkerboard movies (open), and mesopic checkerboard (black) across WT and *Cngb1^neo/neo* retinas. ρ is the linear correlation between the mean information rate and the mean firing rate across retinas. Source files for (**B–H**) are available in *Figure 10—source data 1*.

The online version of this article includes the following source data and figure supplement(s) for figure 10:

**Source data 1.** Contains values for distributions shown in panels B - H.

**Figure supplement 1.** Natural movie information rate changes are not driven by experiment-to-experiment variability.

## Degenerating retina more robustly encodes natural than artificial stimuli

Given the wide range of information rates observed across RGCs for checkerboard stimuli, we hypothesized that the results described above may depend on stimulus choice. Thus, we repeated the experiments under photopic conditions (10,000 Rh*/rod/s) using two natural movies (see 'Materials and methods'). As with the checkerboard stimuli, RGCs again exhibited a wide range of information rates when presented with the natural movies (*Figure 10A*). However, for natural movies, the median information rates were more stable as photoreceptor degeneration progressed (*Figure 10B*: relative to WT, there was only a 5% decline in MI at 4 months; p-value: 0.543). Similarly, the RGCs with the highest information rates (top 10%) did not show the same decline in information rate as observed with the checkerboard stimuli (*Figure 10C*): from WT to 4 months, there was only a 9.5% decline in information rate (p-value: 0.321). ON brisk-sustained cells declined 29% from WT to 4 months (*Figure 10D*; p-value: 0.004). These trends were not produced by experiment-to-experiment variability (*Figure 10—figure supplement 1*).

Finally, we inspected how the information rates depended on the spike rates. Lower information rates could be caused by stimuli producing fewer spikes in degenerating retinas, and stimuli induced fewer spikes at later stages of degeneration, which is suggested by the lower gains observed among the RGCs with space-time-separable RFs (*Figure 6*). First, we examined the information rates for checkerboard stimuli in units of bits/spike instead of bits/s to control for the number of spikes. Under both mesopic and photopic conditions, the bits/spike plots exhibited similar trends to the bits/s plot, indicating that the change was not simply the product of reduced spiking (*Figure 8—figure supplement 1D*, *Figure 9—figure supplement 1D*).

In addition, we compared the information rate to the firing rate for checkerboard stimuli and the natural movies (*Figure 10E*). When comparing information rates across all RGCs at all timepoints, there is a clear correlation between spike rate and information rate for the checkerboard stimuli and natural movies (*Figure 10E*, gray dots). However, at 7 months for checkerboard stimuli (*Figure 10E*, black dots), RGCs with the highest firing rates exhibited suppressed information rates relative to RGCs with similar firing rates from retinas with less degeneration. For natural movies, RGCs from 7-month retinas exhibited much higher information rates across a similar range of firing rates observed in checkerboard stimuli (*Figure 10E*). There was a sharp drop in the information rates for mesopic checkerboard stimuli from WT to 1-month *Cngb1*<sup>neo/neo</sup> animals (10% most informative cells) that was not present under photopic conditions (*Figure 10F*). Under photopic conditions, information transmission was higher among the 10% most informative cells for natural movies than for checkerboard stimuli (*Figure 10F*). These changes in information rates were not strongly correlated with the changes in spike rates across conditions (*Figure 10G*, $\rho$ = 0.13), suggesting that they are not simply a result of changing spike rates.

## Discussion

Determining the extent to which rod dysfunction, degeneration, and death impact cone-mediated visual signaling in the retina is important for diagnosing and treating RP. It is possible that cone-mediated signals in RGCs, the 'output' neurons of the retina, rapidly deteriorate with the loss of rods and concomitant changes in cone morphology. Alternatively, it is conceivable that RGC signaling is robust to photoreceptor degeneration. We have measured cone-mediated responses from thousands of RGCs from *Cngb1*<sup>neo/neo</sup> mice as rod and cone photoreceptors degenerated. We have spanned early timepoints at which most rods remain and there was little to no cone death or changes in cone morphology, to timepoints at which all rods were lost and cones exhibited abnormal morphologies and were beginning to die. We find that despite clear changes in cone morphology and density, RF structure and signal fidelity remained relatively stable until the latest stages of degeneration. In this mouse line, oscillatory spontaneous activity among RGCs did not arise until all or nearly all photoreceptors were lost in contrast to previous results from other RP models. This study highlights how RGC physiology can change or remain robust along different axes, from RFs to contrast response functions to response reliability/information rates. Thus, to get a comprehensive understanding of how RP (and interventions including gene/cell therapy, implants, and optogenetics) impacts visual function, many axes should be considered.

## Comparison to previous studies of RGC signaling in retinitis pigmentosa

Few studies track cone-mediated RGC signaling at many timepoints across RP, in part because in many animal models, the degeneration progresses quite rapidly. Focusing on RGC signaling has the advantage that it includes degradation in visual signaling caused by the degeneration of photoreceptors as well as downstream changes in retinal circuitry. Thus, it captures net changes in retinal processing that deteriorate signaling along with mechanisms that may serve to preserve signaling. Additionally, examining cone-mediated vision instead of rod vision focuses on changes that are most likely to be relevant for humans, given the high reliance on cone-mediated vision. The *Cngb1^{neo/neo}* degeneration rate (relative to the life span of a mouse) is also more similar to the temporal progression of humans with RP, who typically have gradual rod loss over many decades and relative preservation of the cone-dense fovea (*Hull et al., 2017*).

Previous work in a *P23H* rat model of RP (*rho* mutation) indicated a monotonic decrease in RF size, a monotonic increase in the duration of temporal integration, and a rise in spontaneous activity as photoreceptors died (*Sekirnjak et al., 2011*). In contrast, using *Cngb1^{neo/neo}* mice we observed small shifts in RF size and temporal duration during degeneration, but these changes were not monotonic with disease progression. The only features of visual signaling that we found to change monotonically were response gain (*Figures 4 and 6*) and the fraction of RGCs that were light responsive (*Figure 2— figure supplement 1*). While we do not know why non-monotonic changes are occurring for some RF properties, they largely occurred in the 3- to 5-month range. During this time, there is a transient decrease in the rate of rod death (4–5 months) and cone death begins (*Figure 1*). Consequently, there may be complex changes to retinal circuitry as the retina reacts to a temporary stabilization in rod numbers and an acceleration in cone death. Intracellular studies of the light-driven synaptic currents impinging onto bipolar cells and RGCs during this time will be important for understanding the origin of these non-monotonic changes in RF properties.

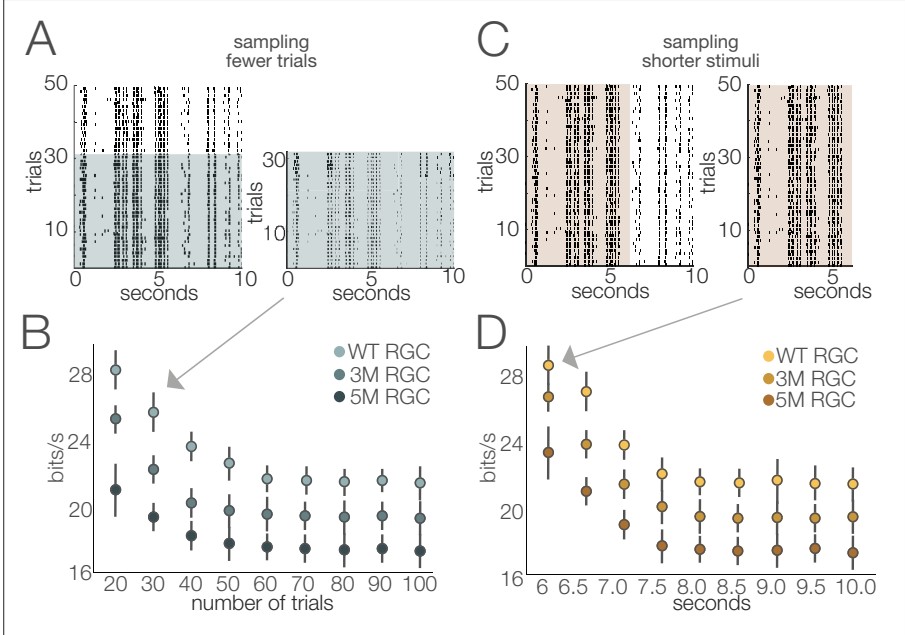

**Figure 11.** Stability of estimated information rates to trial number and trial duration. (**A**) Example rasters illustrating the subsampling for fewer trials. (**B**) Information rates (mean ± 2× SE) of four retinal ganglion cells (RGCs) from separate cohorts as a function of number of repeated stimuli. Information rates stabilized at ~60 trials (repeats). (**C**) Example rasters illustrating the subsampling of trial duration. (**D**) Information rates (mean ± 2× SE) of three RGCs from separate degeneration cohorts by subsampling briefer stimulus durations. The stimulus used in this analysis was movie 1 (see 'Materials and methods'). Similar results were obtained for the other stimuli (movie 2 and checkerboard noise). Source files for (**B**) and (**D**) are available in *Figure 11—source data 1*.

The online version of this article includes the following source data for figure 11:

**Source data 1.** Contains values for panels B and D.

We further speculate that the differences between these studies arise from degeneration having different causes in *P23H* rats versus *Cngb1^neo/neo^* mice. The rat model involves a mutation in rhodopsin, which causes issues with protein folding and trafficking that are not thought to initially change the dark current of rods or their resting glutamate release (*Jones and Marc, 2005*; *Liu et al., 1996*; *Machida et al., 2000*; *Sakami et al., 2011*). However, the *Cngb1^neo/neo^* rods exhibit a sharply reduced dark current, are tonically hyperpolarized, and thus likely release less glutamate onto postsynaptic bipolar cells (*Wang et al., 2019*). These differences in rod physiology and how rods signal to downstream circuits are likely to have consequences for how the retina responds to rod death (see next section). Fully understanding how different origins of rod degeneration impact retinal circuits and RGC signaling is an important direction for future RP studies.

A novel feature of this study is the use of information theory to assay the reliability of visual signaling during retinal degeneration (*Figures 8–10*). Information theory has been used extensively in the vertebrate and invertebrate visual systems, as well as other sensory systems to quantify encoding performance of sensory neurons (*Fairhall et al., 2006*; *Koch et al., 2006*; *Rieke et al., 1995*; *de Ruyter van Steveninck et al., 1997*). In general, sensory neurons with more reliable and selective responses exhibit higher information rates, measured in bits per second or bits per spike. Information theory has not been used previously to assay the effects of neural degeneration on sensory coding. Information theory has the advantage in that it provides a way of comparing the signal fidelity across neurons that transmit information about very different sorts of features (e.g., color vs. motion). It has the potential disadvantages that it is a data-hungry analysis and can be prone to certain biases (*Paninski, 2003*). The relatively long and stable nature of MEA experiments allowed us to overcome these limitations (*Figure 11*). Consistent with previous results (*Koch et al., 2006*; *Koch et al., 2004*), we found a broad range of information rates across the population of RGCs, even in control (WT) retinas. What was perhaps surprising is how stable these information rates were under photopic conditions until those latest stages of degeneration. This indicates that while gain diminishes with degeneration (*Figures 4 and 6*), the precision and reliability of RGC spiking remain relatively stable (*Figures 8–10*). This is because information will depend more on the precision of spiking than on the total number of spikes. For a neuron with Poisson spiking, reliability will go down in lower spike rates, but RGCs exhibit much more reliabe spiking than is consistent with a Poisson process (*Berry et al., 1997*). Interestingly, this reliability of RGC signaling was more robust when using natural stimuli compared to checkerboard noise (*Figure 10*). This difference may be the result of natural stimuli having spatiotemporal correlations that are not present in checkerboard noise, and homeostatic mechanisms that preserve retinal signaling during degeneration can potentially exploit these correlations for higher fidelity signaling.

## Visual compensation for cell loss begins in the retina

Human patients with RP45 (*Cngb1* mutations) maintain cone vision for many years, but the assumption has been that vision was preserved primarily through cortical compensation (*Ferreira et al., 2017*; *Hickmott and Merzenich, 2002*; *Keck et al., 2013*; *Keck et al., 2011*; *Keck et al., 2008*; *Merabet and Pascual-Leone, 2010*; *Turrigiano, 2012*). Indeed, studies in primary visual cortex using the *S334ter* rat model of RP support the notion that cortical plasticity bolsters visual processing during retinal degeneration (*Chen et al., 2016*). However, our findings indicate that retinal signaling remains relatively robust under cone-mediated conditions despite large-scale rod loss, changes in cone morphology, and cone density. This suggests that at least some mechanisms are required to preserve visual signaling during photoreceptor degeneration.

There are two potential classes of mechanisms for this compensation. First, homeostatic plasticity has been documented in models of photoreceptor loss in which the retina remodels to preserve signal transmission (*Care et al., 2019*; *Keck et al., 2013*; *Keck et al., 2011*; *Keck et al., 2008*; *Leinonen et al., 2020*; *Shen et al., 2020*). Alternatively, functional redundancy within the circuit could explain how robust retinal signaling is retained longer than the changes in cone morphology would suggest (*Care et al., 2020*). This study did not distinguish between the two compensation models.

At the latest stages of photoreceptor degeneration in the *Cngb1^neo/neo^* mice (5–7 months), we did observe a decrease in the fraction of RGCs with spike rates that were strongly modulated by checkerboard noise (*Figure 2—figure supplement 1*). It is possible these RGCs were losing their light response completely, or that changes in their light response properties made them relatively unresponsive to checkerboard noise. If the former, it is possible that light-responsive RGCs are becoming

sparser at the later stages of degeneration, which may result in inhomogeneous, or 'patchy,' visual sensitivity described by RP patients (see reviews by *Hull et al., 2017*; *Nassisi et al., 2021*).

## Oscillatory spontaneous activity appears late in *Cngb1^{neo/neo}* mice

A prominent feature of RP models is the emergence of abnormal spontaneous activity among RGCs (*Trenholm and Awatramani, 2015*). Previous studies on other models of RP, particularly *rd1* and *rd10* mouse models of *Pde6b*-RP, have shown abnormal spontaneous activity in two frequency bands: 1–2 Hz and 5–10 Hz (*Biswas et al., 2014*; *Marc et al., 2007*; *Margolis et al., 2008*; *Menzler and Zeck, 2011*; *Poria and Dhingra, 2015*; *Stasheff et al., 2011*; *Tu et al., 2016*; *Ye and Goo, 2007*). Abnormal horizontal cell activity is the source of the 1–2 Hz oscillations (*Haq et al., 2014*), while AII amacrine cells are the likely source of the higher frequency oscillations (*Borowska et al., 2011*; *Choi et al., 2014*; *Ivanova et al., 2016*). Elevated and/or oscillatory spontaneous activity is a cause for concern in retinal degeneration because it is noise that competes with and deteriorates visual signals, particularly near threshold. It is unclear whether RGC oscillations also develop in humans with RP, although reports of phosphenes in RP patients could be explained by spontaneous activity (*Gekeler et al., 2006*).

While previous studies have observed the emergence of oscillations in rodent models of RP (e.g., *rd10*) prior to the loss of photoreceptors and visual signaling, we only found evidence of spontaneous activity following total loss of photoreceptor signals in *Cngb1^{neo/neo}* mice (9 months, *Figure 7*). At 9 months, we could not elicit visual responses from retinas when the video display was switched from 'off' (0 Rh*/rod/s) to 'on' and displaying a 'white' screen (20,000 Rh*/rod/s). Furthermore, at 9 months, there was no clear ONL and cells that labeled for cone arrestin were very sparse, dysmorphic, and did not co-label for M-opsin, indicating near total photoreceptor loss (*Figure 1—figure supplement 1*). Even at 7 months, when nearly all the rods and ~30% of the cones have died, we observed no evidence for changes in spontaneous activity or oscillations.

We suggest that the differences of when oscillations originate may be produced by differences in the resting membrane potential of bipolar cells across different genetic disorders leading to RP. *Pde6b* mutations (*rd1/10*) result in chronically depolarized PRs, which continually release glutamate onto ON bipolar cells. This ultimately keeps the Trpm1 cation channels closed and causes ON bipolar cells to be chronically hyperpolarized (*de la Villa et al., 1995*; *Koike et al., 2010*; *Morgans et al., 2009*; *Slaughter and Miller, 1981*). Rhodopsin mutations produce a similar phenotype. However, photoreceptors lacking Cngb1 are chronically hyperpolarized, resulting in a decrease in glutamate release and chronic depolarization of ON bipolar cells. Eventually, the total loss of photoreceptors results in ON BCs that are chronically hyperpolarized due to downregulation of proteins involved in the metabotropic glutamate receptor cascade, including Trpm1 (*Gayet-Primo and Puthussery, 2015*; *Marc et al., 2007*; *Strettoi et al., 2002*; *Strettoi and Pignatelli, 2000*; *Varela et al., 2003*). Thus, oscillations may arise in amacrine cells (and propagated to RGCs) due to the chronic hyperpolarization of ON bipolar cells, which does not occur until the photoreceptors are lost in *Cngb1^{neo/neo}* but occur much earlier in models such as rd1 and rd10. This is a promising observation from the perspective of gene therapy for rescuing photoreceptors for RP45: it suggests that elevated spontaneous activity and oscillation in RGC spiking will be avoided given that gene therapy is delivered prior to total cell loss.

## Implications for therapy

Our findings on the longevity of cone vision are potentially useful in determining the time window for therapeutic intervention in patients with RP45. Both preclinical and clinical studies have shown that late intervention does not ultimately halt photoreceptor cell death despite improvements to visually guided behaviors (*Bainbridge et al., 2015*; *Cideciyan et al., 2013*; *Gardiner et al., 2020*; *Jacobson et al., 2015*; *Koch et al., 2012*). It is currently unclear whether the continued degeneration is slowed by therapy or not, or what the long-term implications are for vision restoration. This study suggests that cone preservation is a realistic therapeutic target because cone-mediated signaling by RGCs is minimally perturbed by massive rod loss. However, it is not clear that rod-directed gene therapy will be successful at preserving cone (or rod) vision if it is delivered at a timepoint after which most of the rods have died. Thus, an important direction for future work is to determine at what timepoint

rod-directed gene therapies need to be delivered to halt rod death such that cone vision can continue to function normally throughout the remaining life span of the treated individual.

## Materials and methods

### Mice

Mice were used according to Duke University Institutional Animal Care and Use Committee guidelines (protocol A084-21-04) and the Association for Research in Vision and Ophthalmology guidelines for the use of animals in vision research. All mice were housed with 12 hr light/dark cycles and fed rodent chow ad libitum. *Cngb1neo/neo* mice of both sexes were used (12 males and 12 females). The *Cngb1neo/neo* line has a neomycin resistance cassette inserted at intron 19 to disrupt *Cngb1* mRNA splicing (*Chen et al., 2010*; *Wang et al., 2019*). Control animals (WT) consisted of male and female heterozygous littermates between 1- and 9-month postnatal age to account for aging (four males and three females). No age-related differences were found between WT mice (*Figure 2—figure supplement 1*, *Figure 3—figure supplement 1*, *Figure 4—figure supplement 1*, *Figure 5—figure supplement 1*, *Figure 6—figure supplement 1*, *Figure 8—figure supplement 1*, *Figure 9—figure supplement 1*, *Figure 10—figure supplement 1*), so results were pooled. Genotyping was performed by Transnetyx using the primers/probe for the neomycin insert FWD GGGCGCCCGGTTCTT, REV CCTCGTCCTGCA GTTCATTCA, PROBE ACCTGTCCGGTGCCC, and primers/probe for WT *Cngb1* FWD TCCTTAGG CTCTGCTGGAAGA, REV CAGAGGATGAACAAGAGACAGGAA, PROBE CTGAGCTGGGTAATGTC. At least three mice were used per timepoint, except to assay spontaneous activity at 9 months, in which two mice were used (*Figure 7*).

### Immunohistochemistry and confocal microscopy

The samples of retina placed on the MEA and contralateral (unrecorded) eye were fixed for 30 min in 4% PFA (Thermo, 28908) at room temperature. Fixed eyes were hemisected with cornea and lens removed prior to immunolabeling.

For cryosections (*Figure 1B and C*), eye cups were placed in cold 30% sucrose for 3–12 hr, coated in Optimal Cutting Temperature Media (OCT; Tissue-Tek, 4583), placed in a microcentrifuge tube filled with more OCT, frozen using a bath of dry ice and 95% ethanol, and stored at –20°C for at least 24 hr. 12-µm sections were cut using a Leica cryostat (CM3050) and mounted onto frost-free slides (VWR, 48311-703) stored at –20°C. To stain cryosections, slides were warmed to room temperature, rinsed 3× with 1× phosphate-buffered saline (PBS; Santa Cruz, sc-296028), then incubated sequentially with 0.5% TritonX-100 (Sigma, X100) and 1% bovine serum albumin (BSA) (VWR, 0332) for 1 hr each. Primary antibodies were diluted with 0.3% TritonX-100 + 1% BSA, applied to slides at 4°C, and incubated overnight. Slides were rinsed 3× with 1× PBS before applying secondary antibodies diluted with 1× PBS. After incubating at room temperature for 1 hr, slides were again rinsed 3× with 1× PBS, covered with mounting media containing DAPI (Invitrogen, P36935), coverslipped, and sealed with clear nail polish.

For whole mounts (*Figure 1D*), retinal pieces were incubated with 4% normal donkey serum (NDS, Jackson Immuno, C840D36) in 1× PBS for >12 hr. Primary antibodies were diluted in 4% NDS. Tubes were shielded from light and placed on a rocker at 4°C for 7 days. Retinas were rinsed 3× with 1× PBS, then secondary antibody diluted in 1× PBS applied. Tubes were again placed in a 4°C rocker for 1 day, rinsed 3×, mounted onto filter paper, mounting media applied, coverslipped, and sealed with nail polish.

All slides were kept at 4°C until imaged. Z-stack confocal images were taken using a Nikon AR1 microscope using ×20 air and ×60 oil objectives and motorized stage. Images were processed using Fiji software (*Schindelin et al., 2012*). Z-stacks were flattened according to their standard deviation. Brightness and contrast were adjusted as necessary. Cones were manually counted from 60× cross-sections labeled with antibodies to cone arrestin (3–5 images per group). Rod counts were obtained by measuring the area of the ONL and average DAPI nucleus count using the 'Measure' function in Fiji, with cone counts subtracted.

Antibodies used were as follows: rabbit anti-mCar 1:500 (Millipore AB15282, RRID:AB_1163387), mouse anti-PCP2 1:500 (Santa Cruz sc-137064, RRID:AB_2158439), rabbit anti- M-opsin 1:500 (Sigma

AB5405, RRID:AB_177456), and Alexa Fluor secondaries 1:500 (Invitrogen A31571 and A31572, RRID:AB_162542 and AB_162543).

## MEA recordings

Mice were dark-adapted overnight by placing their home cage in a light-shielded box fitted with an air pump for circulation. All dissection procedures the day of the experiment were carried out in complete darkness using infrared converters and cameras. Mice were decapitated, eyes enucleated, and placed into oxygenated room temperature Ames solution (Sigma, A1420) during retinal dissection and vitrectomy as described previously (*Yao et al., 2018*). An ~1 × 2 mm piece from dorsal retina was placed RGC side down on an MEA with either 512 electrodes spaced 60 µm apart, or 519 electrodes with 30 µm spacing (*Field et al., 2010*; *Frechette et al., 2005*; *Ravi et al., 2018*). Oxygenated Ames perfused the retina throughout the experiment at a rate of 6–8 mL/min, heated to 32°C.

## Spike sorting

Raw voltage traces from the MEA were spike sorted using custom software followed by manual curation as described previously (*Field et al., 2007*; *Shlens et al., 2006*). Briefly, spikes were identified by events that crossed a threshold set to 4 standard deviation from the mean voltage. The electrical event 0.5 ms preceding and 1.5 ms following this threshold was extracted from the recording. These events were accumulated on each electrode. Principal components analysis was used to reduce the dimensionality of these signals from 40 to 5 dimensions. Then a water-filling algorithm and expectation maximization were used to fit a mixture of Gaussian models to identify clusters of spikes (*Litke et al., 2004*). Putative cells with a spike rate >0.1 Hz and with <10% contamination estimated from refractory period violations were retained for further analysis.

## Visual stimuli

The image from a gamma-calibrated OLED display (Emagin, SVGA +XL Rev3) was focused onto photoreceptors using an inverted microscope (Nikon, Ti-E) and ×4 objective (Nikon, CFI Super Fluor ×4). Checkerboard stimuli were created and presented using custom MATLAB code. Light from the OLED display was attenuated using neutral density filters. Retina was allowed to settle in darkness for approximately 30 min prior to initiating data collection. Then, spontaneous activity in darkness was recorded for 30 min. The display was then switched to a gray screen emitting ~100 Rh*/rod/s and the retina was left to adapt for 5 min, after which mesopic spontaneous activity was recorded for 30 min. Repeated checkerboard noise was presented 200×, repeating every 10 s. Additionally, non-repeating checkerboard noise was presented for 30 min to estimate spatial and temporal RFs. For all mesopic checkerboard stimuli, the stimulus refreshed every 66 ms and the size of each square was 150 × 150 µm. After mesopic stimuli were presented, a static screen was presented and the NDF filter removed to allow the retina to adapt to a photopic light level (~10,000 Rh*/rod/s). Checkerboard noise repeats (200×, 10 s) were presented, followed by 30 min of nonrepeated checkerboard stimuli. For the photopic checkerboard movies, the stimulus refreshed every 33 ms and each square was 75 × 75 µm. Two 10 s movie clips were presented 100× to estimate the mutual information between RGC signals and naturalistic movies. Movie 1 was a black and white video from a camera attached to a cat walking in the woods (*Betsch et al., 2004*). Movie 2 was a black and white video from a camera carried by a squirrel, depicting fast-moving tree leaves modified from *Freiheit, 2016*.

## Receptive field estimates

RGC responses to checkerboard noise were used to estimate the spatial and temporal components of the STA (*Chichilnisky, 2001*). The STA estimates the spatial and temporal integration of visual signals within the RF of an RGC. For many RGCs, the STA is not space-time-separable, meaning it cannot be expressed as the outer product of a function that depends only on space and a function that depends only on time. This precludes separately analyzing changes in spatial or temporal integration. To identify RGCs with space-time-separable RFs, SVD was performed on each STA. RGCs with STAs that were well-approximated by a rank 1 factorization were kept for further analysis of their spatial and temporal RFs (*Figure 2*): these were cells for which the rank 1 approximation captured >60% of the variance in the STA. For quantifying temporal integration, the time-to-zero crossing was used. This approximates the time-to-peak in the spiking response for a step increment (for ON RGCs) or step decrement (for

OFF RGCs) of light (*Chichilnisky and Kalmar, 2002*). The analysis required that the temporal filter identified by SVD was biphasic because a well-defined zero-crossing was needed to estimate the time-to-zero: 79% of space-time-separable STAs met this criterion.

We computed the static nonlinearity (a.k.a., contrast response function) for each RGC that met the criteria above. This was computed by convolving the spatiotemporal STA with the checkerboard stimulus (*Chichilnisky, 2001*). This resulted in a generator signal for each frame in the stimulus and was used to produce a histogram of observed spike counts for each generator signal. The contrast response function was then estimated by a logistic function described previously (*Ravi et al., 2018*).

Computing the STAs and contrast response functions is equivalent to assuming (or fitting) a linear–nonlinear model to capture the stimulus–response relationship of each cell. To check this assumption, we compared the predicted firing rate (from the linear–nonlinear model) of each RGC to its observed firing rate. The STA and contrast response function were computed from the nonrepeating checkerboard stimulus and used to predict the response to the 10 s repeating checkerboard stimulus (200 repeats). Comparing to the repeating stimulus allowed computing the fraction of explainable variance in the response of the RGC (*Cui et al., 2016*). Across degeneration, we found the linear–nonlinear model accuracy to be consistent in the selected cells (see *Figure 2—figure supplement 1A and B*).

The SNR of the STAs was computed as the ratio between the median intensity values of pixels in the RF center divided by standard deviation of pixel intensity values far from the RF center. 'Far' was defined by fitting the RF center with a two-dimensional Gaussian and taking pixels that were >4 standard deviations away from the center of that Gaussian fit.

## Mutual information

Mutual information (MI) was used to assess the fidelity of RGC signaling. MI measures how much observing the spike train reduces uncertainty about the stimulus. MI was estimated using the 'direct method' (*Strong et al., 1998*). MI between a response and stimulus was computed as

$$I(S; R) = H(R) - H(R|S) \tag{1}$$

where I(S;R) is the mutual information about the stimulus contained in the response. H(R) here is defined as the Shannon entropy of the response distribution (*Shannon, 1948*). This measure estimates the capacity of a neuron to convey information about the stimulus space. H(R|S) is the conditional entropy, a measurement of how noisy neural responses are across repeated trials of an individual stimulus.

$$H(R|S) = P(s) \sum P(r|s) \log_2(r|s) \tag{2}$$

$$H(R) = - \sum P(r) \log_2(r) \tag{3}$$

where P(r) is the probability of a spike count occurring in a pattern across all trials of the stimulus space, while P(r|s) is the probability of observing a response pattern when a stimulus is presented. These probabilities are estimated by measuring the proportion of the observed response patterns at a given epoch of time from all the response patterns across all epochs of time.

Responses from 200 repeated trials of 10 s white noise were used to estimate mutual information across all recorded RGCs. For each RGC, spike trains were binned to a time resolution that achieved the entropy estimates from the Ma upper bound (*Ma, 1981*; *Strong et al., 1998*). The response pattern length (a.k.a., 'word' length) was selected using the same procedure. The mutual information was calculated using the 'direct method' (*Buracas et al., 1998*; *Strong et al., 1998*). Bins that achieved the Ma upper bound ranged from 4 to 6 ms and response patterns ranged from 3 to 6 bins across all RGCs. The mutual information rate was computed as the quotient of the mutual information and the time length of the response pattern. Analysis of trial number and trial duration indicated that both were sufficiently large to produce stable estimates of the information rate (*Figure 11*).

## Light-responsive RGCs

The proportion of RGCs that were light responsive was determined by computing a ratio between the variance and the mean in the peristimulus time histogram from the responses to the natural movie stimuli. The distribution of this ratio in an experiment was bimodal with unresponsive and responsive

RGCs falling in each mode. A threshold was applied to each experiment to exclude the unresponsive RGCs.

## Spontaneous activity and power spectral analysis

The frequency spectra of spontaneous activity were calculated using the fast Fourier transform (FFT) applied to spike times binned at 1 ms from 30 min of spontaneous activity. Spectra were analyzed using a frequency range from 0.1 to 35 Hz. Peaks in the spectra were quantified for every RGC by computing: $abs(b-a)/abs(a-b)$, where $b$ represents the power at a baseline frequency level and $a$ represents the maximum power. The baseline level was estimated by finding the average power in the 0.1–2 Hz range because this range was consistently flat across experiments and cells.

## Statistical tests

Significant changes across all assays were assessed using the two-way Kolmogorov–Smirnov test, a nonparametric test used to determine whether two sets of samples arise from the same distribution (*Massey, 1952*). p-Values were corrected for multiple comparisons by Bonferroni correction. Error bars indicate the range of values within 2 standard errors (SEs) of the mean, which was estimated by bootstrapping the mean 2000 times.

To measure whether differences across timepoints could be produced by other factors (e.g., experiment-to-experiment variability), a parametric linear mixed effects model was used (*Bates et al., 2015*). The mixed effects model accounts for retina-to-retina variability by adding each experiment as a random effect. This procedure permitted making broad-level inferences about the RGC populations without dependence on experimental variability. In addition, the sex of the animal was considered by including it as an interaction term with the degeneration conditions. This step enabled determining whether degeneration conditions were associated with information rates and RF sizes in a sex-independent fashion. The model indicated that conclusions about the impact of degeneration on RGC signaling were insensitive to both sex and experiment-to-experiment variability.

## Acknowledgements

We thank our funding sources: National Institute of Health (NIH) R01 EY027193-01 (GDF, APS, and JC), NIH NEI core grant EY5722, Holland Trice Foundation, Whitehead Foundation, and Research to Prevent Blindness unrestricted grant to Duke University. We also are grateful to Dr. Jon Cafaro and Dr. Suva Roy for technical assistance and Dr. Erika Ellis for discussions.

## Additional information

### Funding

| Funder | Grant reference number | Author |
|---|---|---|
| National Eye Institute | EY027193 | Alapakkam P Sampath<br>Jeannie Chen<br>Greg D Field |
| National Eye Institute | EY5722 | Greg D Field |

The funders had no role in study design, data collection and interpretation, or the decision to submit the work for publication.

### Author contributions

Miranda L Scalabrino, Conceptualization, Data curation, Software, Formal analysis, Supervision, Funding acquisition, Validation, Investigation, Visualization, Methodology, Writing – original draft, Project administration, Writing – review and editing; Mishek Thapa, Conceptualization, Data curation, Software, Formal analysis, Supervision, Validation, Investigation, Visualization, Methodology, Writing – original draft, Writing – review and editing; Lindsey A Chew, Data curation, Software, Formal analysis,

Validation, Investigation, Visualization, Writing – original draft, Writing – review and editing; Esther Zhang, Investigation; Jason Xu, Formal analysis; Alapakkam P Sampath, Conceptualization, Formal analysis, Supervision, Funding acquisition; Jeannie Chen, Conceptualization, Resources, Supervision, Funding acquisition, Validation, Project administration, Writing – review and editing; Greg D Field, Conceptualization, Resources, Data curation, Software, Formal analysis, Supervision, Funding acquisition, Validation, Methodology, Writing – original draft, Project administration, Writing – review and editing

### Author ORCIDs
Miranda L Scalabrino (iD) http://orcid.org/0000-0003-0158-5170
Mishek Thapa (iD) http://orcid.org/0000-0002-2868-7348
Lindsey A Chew (iD) http://orcid.org/0000-0002-2040-1579
Jason Xu (iD) http://orcid.org/0000-0001-5472-3720
Jeannie Chen (iD) http://orcid.org/0000-0002-7904-9629
Greg D Field (iD) http://orcid.org/0000-0001-5942-2679

### Ethics
Mice were used according to Duke University Institutional Animal Care and Use Committee guidelines (protocol A084-21-04) and the Association for Research in Vision and Ophthalmology guidelines for the use of animals in vision research.

### Decision letter and Author response
Decision letter https://doi.org/10.7554/eLife.80271.sa1
Author response https://doi.org/10.7554/eLife.80271.sa2

## Additional files

### Supplementary files
• MDAR checklist

### Data availability
Data to generate all summary plots in Figures 1-11 are included in the following GitHub repository: https://github.com/mishek-thapa/cng-data and are also available as source data files with the manuscript. For physiology data, we have not provided the raw data files (voltage as a function of time on all electrodes) because these files are enormous (in excess of 5 TB). Raw data will be provided upon request by contacting the corresponding author. Requests will be met provided the data will not be used for commercial purposes. MATLAB code for information calculations are available in the above GitHub repository. The Cngb$^{neo/neo}$ mouse model is available to be shared upon request. Raw image files from Figure 1 can be found at https://doi.org/10.5061/dryad.x95x69pmq.

The following dataset was generated:

| Author(s) | Year | Dataset title | Dataset URL | Database and Identifier |
|---|---|---|---|---|
| Scalabrino ML | 2022 | Robust cone-mediated signaling persists late into rod photoreceptor degeneration | https://doi.org/10.5061/dryad.x95x69pmq | Dryad Digital Repository, 10.5061/dryad.x95x69pmq |

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
