## [Editor Report]

This is an important study describing the decline of retinal function in a mouse model of slow photoreceptor degeneration. The authors present compelling evidence based on a characterization of functional changes across some RGC populations and information theory to assess the fidelity of the remaining. They show remarkable preservation of cone-driven ganglion cell light responses in advanced stages of a retinitis pigmentosa model when most rods have died and cone morphologies are dramatically altered. The results are presented clearly in the text and figures and are discussed in the context of previous studies on retinal degeneration.

---

## [Decision Letter]

**Decision letter after peer review:**

Thank you for submitting your article "Robust cone-mediated signaling persists late into rod photoreceptor degeneration" for consideration by *eLife*. Your article has been reviewed by 4 peer reviewers, one of whom is a member of our Board of Reviewing Editors, and the evaluation has been overseen by Tirin Moore as the Senior Editor. The reviewers have opted to remain anonymous.

Essential revisions:

There are 4 reviewers all with mixed enthusiasm. Everyone appears to think that the observation that even without most rods and with quite abnormal cone morphologies the retinal output under photopic conditions is relatively well preserved is an impactful result. And the reviewers mostly are satisfied with the quality of the data and the presentation. However, all reviewers had conceptual concerns and some technical concerns. It is hard to summarize them entirely but this is what I glean to be essential. Note this is not a complete list – you are highly encouraged to go through the comments on each of the reviewers.

1. All reviewers identify the authors' claim that they have characterized the "retina as a whole" as a major weakness of the study. There are several suggestions by the various reviewers that the authors should consider. These include:

– Some characterization of the number of cells that retain their light responsiveness. The primary quantification of this is in Figures 2E/F. How this is just the fraction of RGCs that have separable S-T receptive fields NOT the fraction that is light responsive. As noted by Reviewer #2 this feels like a fundamental measure that must be included to assess the visual responsiveness of the 'retina as a whole".

– Select a cell type to follow over time. Some reviewers suggest direction-selective cells since they are readily classified and the authors have a history of readily recording these cell types. However, the authors have made clear that they want to stick to cells with separable space-time receptive fields, in which case perhaps they can select say the brisk transient cell.

2. There is a technical concern regarding the very low repeat reliability the authors show even for RGC responses in wild-type retinas indicating that the vast majority of RGC spike trains contain little information about naturalistic or white noise stimuli. As noted, if repeat reliability is so low, it is unlikely that the LN models predict the responses well. Perhaps you can show what fraction of the variance their LN models explain. If the models are not a good description of the RGC responses in their recordings, then using the model parameters to compare responses across genotypes and time is suspect. Moreover, Inter-experimental variation is well documented in retinal recordings (incl. MEA). Because one can get many cells in an individual recording and cell #s can be unevenly distributed across recordings, it is important to confirm that key trends hold when n = experiments rather than n = cells.

3. There were several specific questions regarding the implementation of information theory approaches. This is potentially the general result of the study and therefore should be strengthened.

4. All reviewers have made comments regarding the conceptual advance. Reviewer 4 noted that the evidence of homeostatic compensation is limited. Reviewer 2 asked for more discussion regarding how this particular model of photoreceptor degeneration compares to the many others that have been studied. Reviewer 4 states that the interpretation provided by the authors that homeostatic plasticity preserves information encoding in the retinal output is rather weak since the study does not actually show any homeostatic changes in circuits (unlike other studies on this topic).

*Reviewer #1 (Recommendations for the authors):*

Figure 3

Can the authors provide an explanation for what happens at 4M?

Legend is missing an explanation of gray/dark bars and teal diamonds.

Figures 3 and 5

Would the point about variance be clearer with a quantification of variance or change in variance over time?

Figure 7: Is the figure missing a significance asterisk for panel G?

Figure 9 legend: reference to lines is incorrect.

Line 231:

Can the authors provide more explanation for why there would be a change in SNR but only at one time point?

Line 255

What is the explanation for how photopic RGC signaling is maintained with lower gain and lower SNR if oscillations cannot provide the explanation?

Line 386

Why does this model not have oscillatory spontaneous activity until 9M while the other RP models have oscillatory activity? Is the difference associated with timing?

---

## [Author Response]

Essential revisions:1. All reviewers identify the authors' claim that they have characterized the "retina as a whole" as a major weakness of the study. There are several suggestions by the various reviewers that the authors should consider. These include:– Some characterization of the number of cells that retain their light responsiveness. The primary quantification of this is in Figures 2E/F. How this is just the fraction of RGCs that have separable S-T receptive fields NOT the fraction that is light responsive. As noted by Reviewer #2 this feels like a fundamental measure that must be included to assess the visual responsiveness of the 'retina as a whole".

We thank the reviewers for their constructive feedback and enthusiasm for the significance of the study. For clarity, new text in the manuscript is blue.

We now include a quantification of the fraction of RGCs in our experiments that retain light responses as a function of degeneration. This is in Figure 2—figure supplement 1. The number of RGCs measured on the MEA was given by the number of spike clusters that exhibited a refractory period (3ms after the spike) with less than 10% contamination and a spike rate of greater than 0.1Hz. The number of those RGCs with visually driven responses was estimated from the checkerboard noise stimulus by computing the ratio between the mean spike rate and the variance: spike rates modulated by the stimulus will exhibit a high variance to mean ratio. We set this threshold at a range of 0.9 -1.2 sec/spike depending on the experiment. Under photopic conditions, this fraction did not change until 5M and fell to zero at 9M. (Figure 2—figure supplement 1C-D and lines 167-174)

We regret generating the impression that we were characterizing the “retina as a whole”. In fact, we never use that phrase. We do use the phrase quantifying the “net impact” of photoreceptor degeneration on RGC output, meaning a measurement that nets the impact of changes in the photoreceptors with changes in retinal circuits. We believe our study accomplishes this for those RFs that are approximately linear and space-time separable as well as for information transmission. We have refined and clarified our language on this point at Lines 63-66.

– Select a cell type to follow over time. Some reviewers suggest direction-selective cells since they are readily classified and the authors have a history of readily recording these cell types. However, the authors have made clear that they want to stick to cells with separable space-time receptive fields, in which case perhaps they can select say the brisk transient cell.

As suggested by the reviewers, we have added an analysis of a specific RGC type: ON brisk sustained RGCs. Their photopic and mesopic receptive fields and information rates are tracked throughout the manuscript (Figures 3-6 and 8-10). Example receptive field mosaics, temporal receptive fields, and spike train autocorrelation functions for WT and 4M Cngb1^neo/neo^ animals are shown in Figure 2—figure supplement 1E-F. These RGCs follow the trends displayed by the larger population of RGCs in each analysis. We chose this cell type because they are readily identified by their spike train autocorrelation functions compared to other RGC types and they have approximately space-time separable receptive fields (RFs). There are many text changes associated with adding an analysis of the ON Brisk sustained RGCs (see lines 202-207; 227-229; 264-267, etc).

We chose not to focus on direction selective RGCs because we are analyzing the spatial and temporal RFs of RGCs in Figures 3-5 and direction-selective RGCs do not have space-time separable RFs (see example in Figure 2C-D). Thus, those cells could not be used to track those receptive field properties across degeneration. Also, we did not collect responses to drifting gratings or bar responses across a range of speeds or contrasts, so we are unable to reliably distinguish the different types of direction-selective RGCs (e.g., ON vs ON-OFF) from these data.

2. There is a technical concern regarding the very low repeat reliability the authors show even for RGC responses in wild-type retinas indicating that the vast majority of RGC spike trains contain little information about naturalistic or white noise stimuli. As noted, if repeat reliability is so low, it is unlikely that the LN models predict the responses well. Perhaps you can show what fraction of the variance their LN models explain. If the models are not a good description of the RGC responses in their recordings, then using the model parameters to compare responses across genotypes and time is suspect. Moreover, Inter-experimental variation is well documented in retinal recordings (incl. MEA). Because one can get many cells in an individual recording and cell #s can be unevenly distributed across recordings, it is important to confirm that key trends hold when n = experiments rather than n = cells.

The first question raised here is regarding the repeatability of the responses to checkerboard noise and natural movies shown in Figures 8-10: some RGCs exhibit high repeatability, some exhibit low repeatability as quantified by their information rates. The reviewers are concerned about those cells with low repeatability and the ability of capturing their responses with an LN model. This is a valid concern, but to be clear, we are NOT fitting an LN model to cells with low information rates. Note, in Figures 3-6, where an LN model is being used to estimate the spatial and temporal components of the RFs, we are fitting a subset of all the RGCs: those with space-time separable RFs (see Figure 2). Those particular cells exhibit high information rates and highly reproducible responses, and an LN model captures 55-60% of the variance in the spike rate. We have added Figure 2—figure supplement 1A-B to show that the LN model is capturing their responses reasonably well. We explain this in the Methods (lines 717-726) and Results (lines 147-151).

In addition, the information rates we estimated in mouse are consistent with past studies from guinea pig (Koch et al., 2004 and Koch et al., 2006). We have updated the example neurons to better reflect the reliability of the cells near the median of the MI distributions in Figures 8-10.

The second point raised here is with regards to analyzing experiment-to-experiment variability and the extent to which that could contribute to effects that we observed. We have added many supplemental figures (see the figure supplements to Figures 3, 4, 5, 6, 8, 9 and 10) that show the per-experiment effects. None of the trends we described or highlighted in this study were caused by experiment-to-experiment variability. We also make this point at several locations in the Results (e.g. lines 208-211; 228-229; 266-268, etc).

3. There were several specific questions regarding the implementation of information theory approaches. This is potentially the general result of the study and therefore should be strengthened.

We handle these as they arise for each individual reviewer.

4. All reviewers have made comments regarding the conceptual advance. Reviewer 4 noted that the evidence of homeostatic compensation is limited. Reviewer 2 asked for more discussion regarding how this particular model of photoreceptor degeneration compares to the many others that have been studied.

We now have a subsection that compares RP models (Discussion – line 435): Comparison to previous studies of RGC signaling in retinitis pigmentosa. We think this is a reasonably comprehensive comparison to other studies. If there are specific animal models we are missing that the reviewer(s) would like to see us discuss, please indicate those models.

Reviewer 4 states that the interpretation provided by the authors that homeostatic plasticity preserves information encoding in the retinal output is rather weak since the study does not actually show any homeostatic changes in circuits (unlike other studies on this topic).

We agree with Reviewer 4 that this is a possible alternative. We discuss this alternative in the Discussion section (lines 514-520).

Reviewer #1 (Recommendations for the authors):Figure 3Can the authors provide an explanation for what happens at 4M?

We added additional text at lines 455-462:

“While we do not know why non-monotonic changes are occurring for some RF properties, they largely occurred in the 3-5M range. During this time, there is a transient decrease in the rate of rod death (4-5M) and cone death begins (Figure 1). Consequently, there may be complex changes to retinal circuitry as the retina reacts to a temporary stabilization in rod numbers and an acceleration in cone death. Intracellular studies of the light-driven synaptic currents impinging onto bipolar cells and RGCs during this time will be important for understanding the origin of these non-monotonic changes in RF properties.”

Legend is missing an explanation of gray/dark bars and teal diamonds.

Thank you. We have added legends to each figure indicating what the gray bars signify, and we identify what the diamonds mark in each figure caption.

Figures 3 and 5Would the point about variance be clearer with a quantification of variance or change in variance over time?

Perhaps, but we think the variance is reasonably clearly represented by showing the full distributions as well as the gray bars indicating the (50% and 80% intervals) below the distributions.

Figure 7: Is the figure missing a significance asterisk for panel G?

Thank you for catching this. We have corrected in the updated Figure 7.

Figure 9 legend: reference to lines is incorrect.

Thank you for alerting us to this mistake. The Figure 9 legend has been corrected to change “teal” to “orange”.

Line 231:Can the authors provide more explanation for why there would be a change in SNR but only at one time point?

The SNR of the STA is going to be related to the precision of RGC firing (but less sensitive than measuring information because the SNR averages over many stimuli). So long as the response precision remains high, the SNR of the STA would also be expected to remain high provided that the gain hasn’t dropped to a very low value. The largest decrease in information (reflecting a decrease in spiking precision) occurs at 7M (Figure 9C), and hence we think this is driving the large change in the SNR of the STAs.

Line 255What is the explanation for how photopic RGC signaling is maintained with lower gain and lower SNR if oscillations cannot provide the explanation?

If we understand the Reviewer’s question, the answer is the same as for the question immediately preceding this one. The SNR of the STA is (to a large degree) tracking the precision of spiking (e.g. the information rate for checkerboard stimuli). The information rates were even more robust for natural movies, but we are not measuring the SNR for that stimulus and stimuli with very distinct statistics can drive very different responses in the RGCs.

Line 386Why does this model not have oscillatory spontaneous activity until 9M while the other RP models have oscillatory activity? Is the difference associated with timing?

We have added text to the discussion to expand on oscillatory activity (Lines 555-570). In summary, we hypothesize the onset of oscillatory activity is due to the resting potential of ON bipolar cells. Briefly, Cngb1^neo/neo^ and rd1/10 rods have different resting membrane potential during degeneration (hyperpolarized and depolarized respectively). This likely produces different resting membrane potentials in the rod bipolar cells because of differences in rod glutamate release. We suggest these differences may initiate oscillations at different timepoints during the degeneration process.